# Remodelling of the immune landscape by IFNγ counteracts IFNγ-dependent tumour escape in mouse tumour models

Vivian W. C. Lau [1], Gracie J. Mead [1], Zofia Varyova[1], Julie M. Mazet [1], Anagha Krishnan[1,2], Edward W. Roberts [3], Gennaro Prota[4,5], Uzi Gileadi [4], Kim S. Midwood[1], Vincenzo Cerundolo[4,6] & Audrey Gérard [1] ✉

Loss of IFNγ-sensitivity by tumours is thought to be a mechanism enabling evasion, but recent studies suggest that IFNγ-resistant tumours can be sensitised for immunotherapy, yet the underlying mechanism remains unclear. Here, we show that IFNγ receptor-deficient B16-F10 mouse melanoma tumours are controlled as efficiently as WT tumours despite their lower MHC class I expression. Mechanistically, IFNγ receptor deletion in B16-F10 tumours increases IFNγ availability, triggering a remodelling of the immune landscape characterised by inflammatory monocyte infiltration and the generation of 'mono-macs'. This altered myeloid compartment synergises with an increase in antigen-specific CD8⁺ T cells to promote anti-tumour immunity against IFNγ receptor-deficient tumours, with such an immune crosstalk observed around blood vessels. Importantly, analysis of transcriptomic datasets suggests that similar immune remodelling occurs in human tumours carrying mutations in the IFNγ pathway. Our work thus serves mechanistic insight for the crosstalk between tumour IFNγ resistance and anti-tumour immunity, and implicates this regulation for future cancer therapy.

Tumour escape is a mechanism whereby tumours acquire genetic mutations resulting in the evasion of immunosurveillance. Immune evasion can occur during primary or acquired resistance and is associated with a lack of therapeutic response and subsequent disease progression. Establishment of resistance generally involves loss of T cell-dependent cytotoxicity, which can occur through deficiencies in antigen presentation mechanisms, or acquisition of resistance against interferon gamma (IFNγ)[1,2].

IFNγ induces anti-tumour immunity by exerting direct cytotoxic and cytostatic effects on tumours[3,4], inducing major histocompatibility complex (MHC) expression[5], and promoting the expansion of effector lymphocytes and maturation of myeloid populations[6,7]. But it can simultaneously impede anti-tumour immunity, for example,

through induction of PD-L1 and IDO expression[8,9] and limiting stem-like T-cell driven immunity[10]. As such, IFNγ affects both the tumour itself and its microenvironment, with contrasting consequences on the anti-tumour response[11]. In patients, it is well-established that durable responses to immune checkpoint blockade (ICB) are associated with IFNγ-related gene signatures[12], which are also correlated with increased tumour mutational burden (TMB)[13] and T cell infiltration[14], suggesting that IFNγ-driven responses is a pre-requisite to ICB response. IFNγ induces a complex network of downstream effects mediated through its cognate receptor, IFNγR. IFNγ signalling regulates the expression of hundreds of genes, known as interferon-stimulated genes (ISGs), and interestingly, the balance between immune and cancer ISGs correlates with response to ICB[15],

[1]The Kennedy Institute of Rheumatology, University of Oxford, Oxford, UK. [2]Immunodynamics Section, Cancer and Inflammation Program, Center for Cancer Research, National Cancer Institute, Bethesda, MD, USA. [3]CRUK Beatson Institute, Glasgow, UK. [4]MRC Translational Immune Discovery Unit, John Radcliffe Hospital, University of Oxford, Oxford, UK. [5]Present address: Department of Biomedicine, University of Basel, Basel, Switzerland. [6]Deceased: Vincenzo Cerundolo. ✉e-mail: Audrey.gerard@kennedy.ox.ac.uk

highlighting the importance in eliciting the correct equilibrium between pro- and anti-tumoural IFNγ functions.

Many clinical reports have associated acquired resistance to ICB with loss of IFNγ response by tumour cells via signalling pathway mutations[16–18]. Yet, mutations in IFNγ signalling pathway are relatively infrequent (i.e. <10% patients) in colorectal cancers[19], and *JAK1* mutations were associated with better 5-year survival rates amongst colorectal cancer patients[20]. In patients with frequent mutations such as endometrial cancers, *JAK1* mutations appeared to have little impact on ICB outcomes[21]. Recent meta-analyses focused on leveraging the statistical power of independent observations have discovered that pre-existing IFNγ pathway mutations in multiple cancer types did not necessitate lack of response to ICB[22]. Several in vivo pre-clinical models and CRISPR screens support this observation, as cell lines with mutations in IFNγR or downstream signalling molecules such as JAK1/2 and STAT1 have been shown to sensitise the tumour towards improved ICB response[23,24]. It is therefore important to understand the factors that contribute to eliciting a potent anti-tumour response during tumour escape and how it integrates with mutations in the IFNγ pathway.

IFNγ signatures pre- and post-ICB treatment are associated with clinical response to therapy, and it was often assumed that the main mode of action of IFNγ is to directly inhibit and/or kill tumour cells. As the above studies challenge this dogma, it is still poorly understood mechanistically what controls tumours insensitive to IFNγ.

Here, we study the consequences of deleting the IFNγR in murine melanoma tumour cells on the remodelling of the tumour immune landscape. We found that loss of IFNγR1 on tumour cells results in intra-tumoural IFNγ accumulation, which induces pro-inflammatory signalling of immune-infiltrating populations. The myeloid compartment exhibits substantial immune remodelling in IFNγR-deficient tumours compared to wild-type, through increased recruitment and retention of pro-inflammatory monocytes and decreased immuno-suppressive macrophage generation. More importantly, loss of monocyte infiltration subverts the inflammatory phenotype in IFNγR-deficient tumours, and monocyte intra-tumoural co-localisation with CD8⁺ T cells around blood vessels appears to support their anti-tumour functions. Overall, our study demonstrates that tumour-derived mutations in the IFNγ pathway can trigger a remodelling of the immune landscape underlying the control of those mutated clones. As we also observe this immune landscape remodelling in humans, it highlights the relevance of our findings and provides potential new therapeutic avenues, which could be important beyond IFNγ-insensitive tumours.

## Results

### Loss of IFNγ signalling does not result in decreased patient survival for multiple cancer types

While several reports of IFNγ pathway somatic mutations were found in post-ICB treatment patients, it is often unclear whether those mutations can be present before ICB, and if they affect overall survival of cancer patients. We investigated the prevalence of those mutations before ICB by collating data from The Cancer Genome Atlas (TCGA) for pre-treatment tumours which harboured mutations in the IFNγ receptor subunits (*IFNGR1/2*) or downstream signalling molecules (*JAK1/2, STAT1*). The prevalence of mutations was found to be relatively infrequent (i.e. alteration frequency <10%) across most tumour types except endometrial cancer and melanoma (Fig. 1A), compared to known oncogenic mutations such as *KRAS* or *PIK3CA* (Supplementary Fig. 1A, B). IFNγ pathway mutations were found at similar frequencies compared to antigen presentation mutations such as in *B2M*, *TAP*, or *HLA* molecules (Supplementary Fig. 1C), suggesting that the pressure induced by T cells is not potent enough in most tumours to select for these types of mutations. More importantly, presence of IFNγ-pathway mutations did not result in significantly higher overall mortality in

cancer types with the highest mutational frequency, and even correlated with improved survival in endometrial cancer (Fig. 1B–E). This, combined with previous pre-clinical data which showed enhancement of checkpoint blockade responses in pathway mutants, suggests that tumours with mutations in the IFNγ pathway are as efficiently controlled by the immune system as tumours that are sensitive to IFNγ. This led us to investigate the immunological changes in the tumour microenvironment which may be promoting immunity towards these types of tumours.

### Control of B16F10 IFNγRKO tumours is CD8⁺ T cell-dependent

We established an IFNγ-insensitive model of B16F10 melanoma (Fig. 2A, B) through CRISPR-Cas9 knockout of IFNγR1 (IFNγRKO). As expected, deletion of IFNγR1 results in lack of MHC class I and II upregulation in vivo (Fig. 2C, D), and following IFNγ stimulation in vitro (Supplementary Fig. 2A). This cell line also expresses ovalbumin (OVA) (B16-OVA) to track antigen/tumour-specific T cells. Given that not all individual cancer cells in human tumours are unresponsive to IFNγ, we used a B16-OVA WT and IFNγRKO admix model whereby tumour cells are tagged in ZsGreen or mCherry, or vice versa, and mixed in equal proportions prior to engraftment (Fig. 2E). As a control, we admixed ZsGreen and mCherry WT cells (WT:WT) or ZsGreen and mCherry IFNγRKO (KO:KO) cancer cells. No differences in tumour volumes between control WT:WT, KO:KO or admixed WT:KO tumours were observed in vivo (Fig. 2F and Supplementary Fig. 2B). However, a phenotype emerged whereby IFNγRKO cells outgrew WT cells in a time-dependent matter (Fig. 2G and Supplementary Fig. 2C), which we hereafter refer to as selection. This is consistent with human data suggesting that mutations in the IFNγ pathway do not confer any advantage in tumour growth as a whole or survival, but these mutations can rise and take over other clones. To ascertain whether this selection was dependent on IFNγ-signalling by B16 tumour cells, we created B16-OVA cell line expressing a IFNγR1 mutated on Y445A, which abolishes the STAT1 binding site[25]. Similar to complete IFNγR deletion, Y445A mutation resulted in selection, with mutated cells taking over their WT counterparts (Supplementary Fig. 2D). Furthermore, the selection was lost when admixed tumours were engrafted in IFNγKO mice (Fig. 2H), confirming that selection was induced by IFNγR deletion rather than off-target effects. NK cells have been implicated in the control of tumours with low MHC-I expression[26]. To test whether NK cells were responsible for controlling IFNγRKO tumours, we treated mice with NK1.1 antibodies to deplete NK cells prior to engraftment of WT or IFNγRKO B16-OVA tumours. Consistent with an early role of NK cells in controlling tumour growth[27], NK depletion led to increased tumour growth and decreased survival (Supplementary Fig. 2E, F). However, IFNγRKO tumour volumes were surprisingly equivalent to that of WT tumours when NK cells were depleted, suggesting that NK cells did not provide enhanced control of tumours with low MHC-I and that other immune cells might be involved in controlling IFNγ-insensitive tumours. B16-OVA IFNγRKO tumours are predicted to be less sensitive to CD8⁺ T cells because of their low expression of MHC-I[28,29]. To test this, we engrafted either WT, IFNγRKO or admixed B16-OVA in CD8αKO mice, which are devoid of CD8⁺ T cells. Selection of IFNγRKO tumour cells was not observed in CD8αKO mice; instead, in approximately half the mice, the phenotype was reversed (Fig. 2I). To characterise how CD8⁺ T cells and IFNγ led to the selection of IFNγRKO tumour cells in our admix setup, we first analysed whether IFNγ had a direct cytostatic effect on WT tumours. To do so, we admixed WT and IFNγRKO B16-OVA tumour cells in vitro and added IFNγ to the culture. The proliferation of IFNγRKO tumour cells, as assessed by KI67 staining, was greater than the one of WT tumour cells (Fig. 2J), demonstrating that IFNγ inhibits tumour cell proliferation, which contributed to the selection of IFNγRKO over WT tumour cells (Fig. 2K). To test whether CD8⁺ T cells could also induce IFNγRKO tumour cell selection through preferential killing of WT tumour cells,

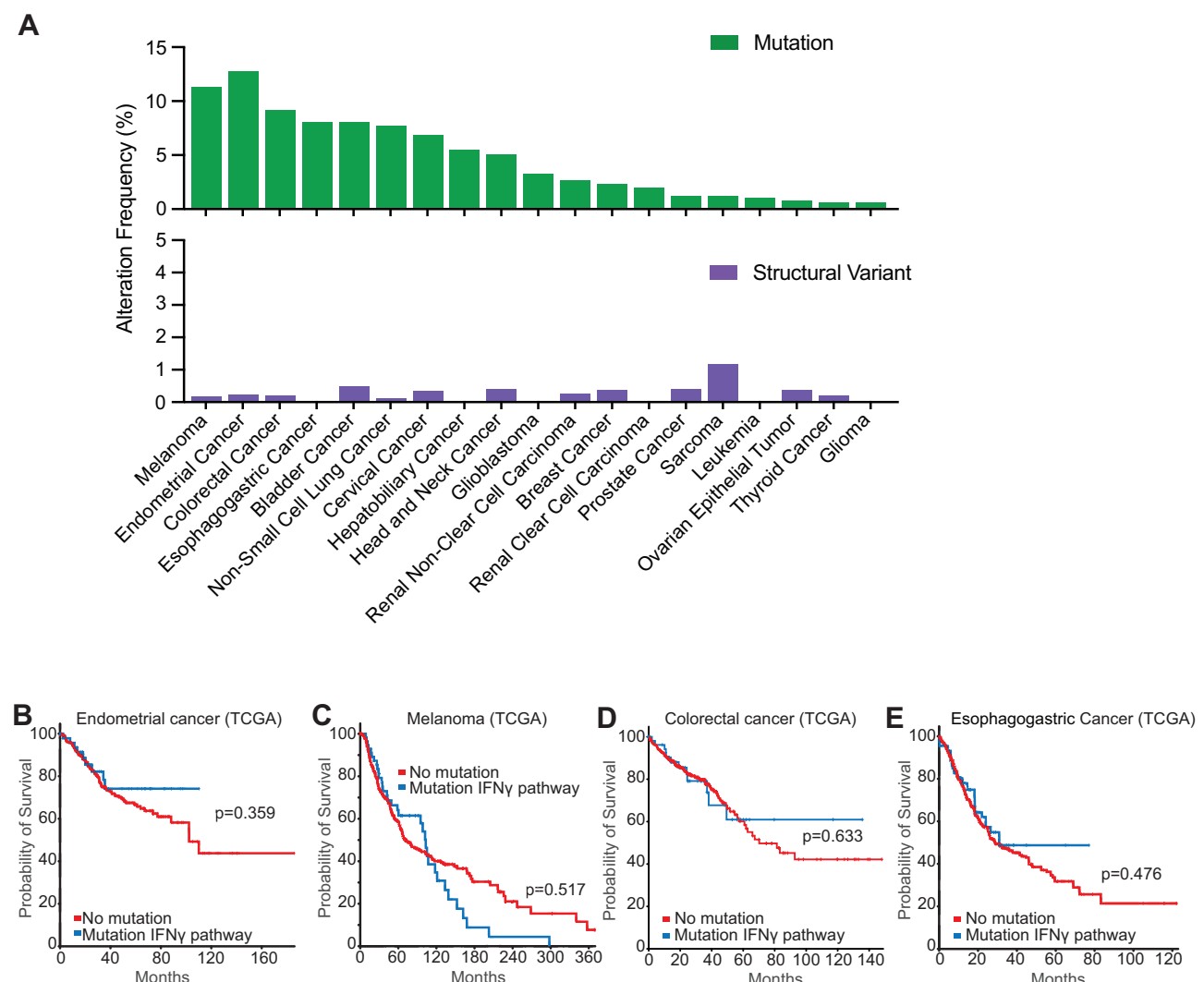

**Fig. 1 | Presence of mutations in IFNγ signalling pathway genes does not preclude decrease in overall survival in clinical data. A** Frequency of alterations in *IFNGR1*, *IFNGR2*, *JAK1*, *JAK2*, or *STAT1* (IFNγ pathway) across cancers in The Cancer Genome Atlas (TCGA), where cases in green represent gene mutations and purple are structural variants of the genes. For endometrial cancer, samples with POLE mutations have been excluded. Comparison of survival curves of endometrial (**B**, $n = 462$ without and $n = 50$ with mutation in the IFNγ pathway, exclusion of samples with POLE mutations), melanoma (**C**, $n = 366$ without and $n = 57$ with mutation in the IFNγ pathway), colorectal (**D**, $n = 473$ without and $n = 49$ with mutation in the IFNγ pathway), and esophagogastric (**E**, $n = 1034$ without and $n = 103$ with mutation in the IFNγ pathway) human cancers between unaltered cases and cases with mutations in IFNγ pathway genes. $p$ values represent log-rank testing.

we admixed WT and IFNγRKO tumour cells and analysed cell death induced by OVA-specific OTI CD8$^+$ T cells using Annexin V staining. We observed a slight but consistent preferential killing of WT over IFNγRKO tumour cells by OTI CD8$^+$ T cells (Fig. 2L), consistent with the increase of MHC-I observed in the WT but not IFNγRKO tumour cells following IFNγ treatment (Fig. 2C). Overall, we concluded that a direct effect of IFNγ on proliferation and indirect effect on MHC-I expression and subsequent killing by CD8$^+$ T cells contribute to the selection of WT over IFNγRKO tumour cells in our admix model.

While CD8$^+$ T cells are required for selection of IFNγRKO over WT tumours, we unexpectedly found tumour growth to be significantly higher in all tumour types when tumours are engrafted in CD8αKO mice (Fig. 2M and Supplementary Fig. 2G), demonstrating that CD8$^+$ T cells are still crucial for controlling IFNγRKO tumours. Consistent with a specific involvement of CD8$^+$ T cells in controlling the growth of IFNγRKO tumours, quantification of lymphocyte tumour infiltration by flow cytometry revealed no overt differences except for an increase in OVA-specific CD8$^+$ T cells in IFNγRKO and admix tumours (Fig. 2N and Supplementary Fig. 2H), as assessed by

OVA-tetramer staining. Interestingly, OVA-specific CD8$^+$ T cell infiltration is similar between WT and tumours that do not express MHC-I (H2-K$^b$/D$^b$KO) (Supplementary Fig. 2I, J), highlighting the importance of the tumour microenvironment for recruiting/maintaining antigen-specific CD8$^+$ T cells at the tumour site. To confirm that CD8$^+$ T cells were still primed by their TCR in H2-K$^b$/D$^b$ KO tumours, we used the Nur77-GFP reporter mice, whereby TCR, but not cytokine triggering induces GFP expression in T cells[30]. OVA-specific CD8$^+$ T cells expressed GFP to a greater extent than the tetramer-negative CD8$^+$ T cells in both WT and H2-K$^b$/D$^b$ KO tumours (Supplementary Fig. 2K), confirming that the microenvironment is critically important to support CD8$^+$ T cells priming and most likely their maintenance/expansion. Finally, increased antigen-specific CD8$^+$ T cell infiltration following IFNγR depletion has been observed in other model antigens and cell lines[15,31], suggesting that this is not an artefact of OVA-expressing tumours.

Thus, despite low MHC-I expression and insensitivity to IFNγ, the control of growth and selection of IFNγRKO tumours remained CD8$^+$ T cell- and IFNγ-dependent.

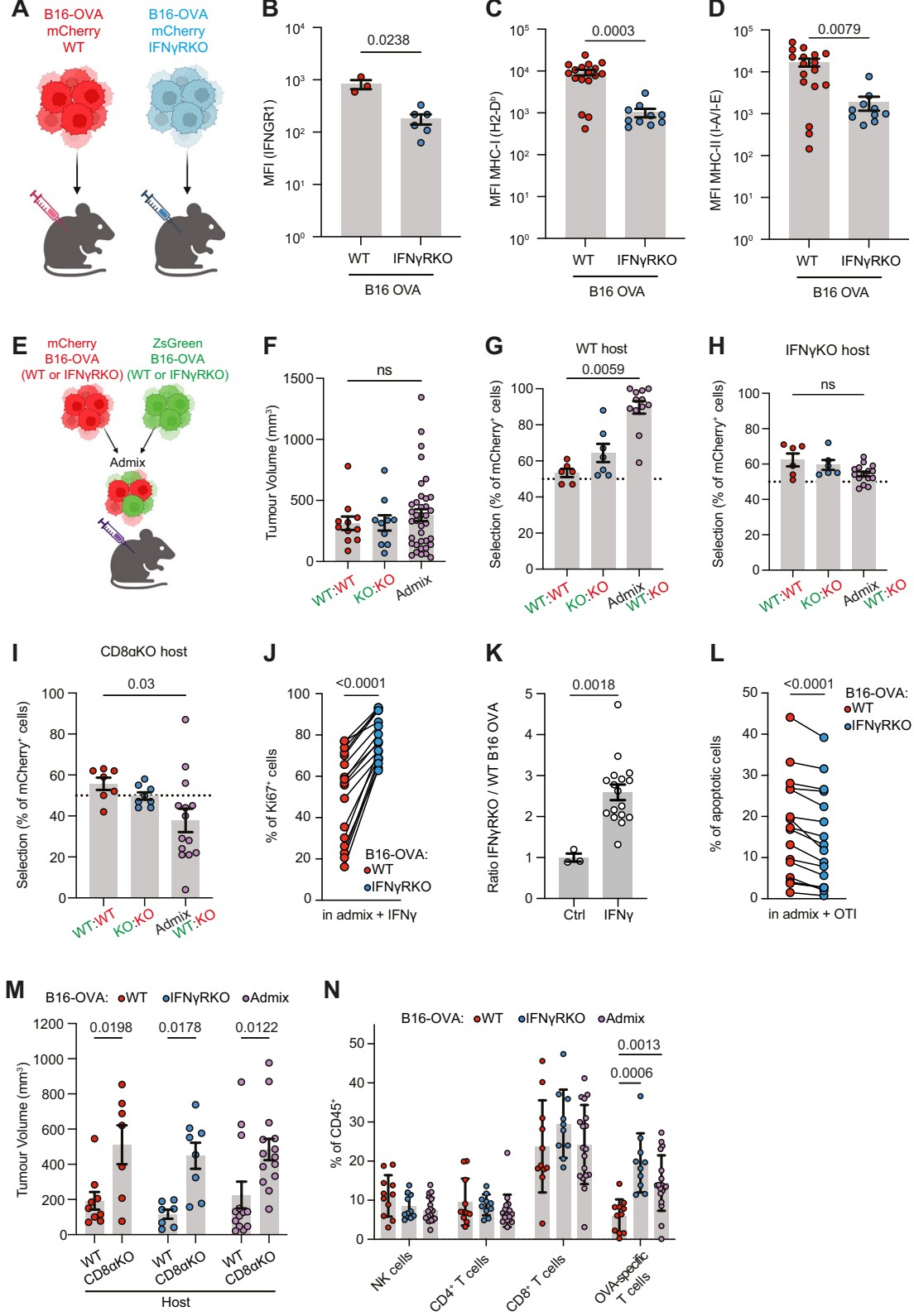

## scRNAseq identifies significant changes in cytokine signalling and myeloid infiltration of IFNγRKO tumours

Increase in tumour-specific CD8[+] T cells suggests that inhibition of IFNγ signalling in tumours might alter the cytokine environment, inducing global changes in signalling pathways of other immune subsets which we explored using genomic methods. We employed single-cell RNA sequencing (scRNAseq) of the CD45[+] compartment

isolated from pooled tumour samples to better understand the immunological changes which enable effective control of IFNγ-insensitive tumours. Clusters were generated using unsupervised hierarchical clustering and annotated using canonical gene expression patterns (Fig. 3A and Supplementary Fig. 3A). We identified six myeloid populations. Of those, we found four dendritic cell (DC) subsets: cDC1 (*Xcr1*), cDC2 (*Cd209a, Clec10a*), mReg DCs (*Cd200,*

**Fig. 2 | B16F10 melanoma tumours with CRISPR-Cas9 knockout of IFNγR1 are efficiently controlled by the endogenous anti-tumour response. A–D** B16-OVA WT (red) or IFNγRKO (blue) cells were engrafted subcutaneously into the flanks of C57Bl/6 WT mice and tumours were harvested after 11–16 days. **A** Diagram of the experimental setup. Graphics created with BioRender. Surface expression of IFNγR1 (**B** $n_{[WT]} = 3$, $n_{[IFNγRKO]} = 6$ for from two independent experiments), MHC-I H2-D$^b$ (**C** $n_{[WT]} = 17$, $n_{[IFNγRKO]} = 10$ from four independent experiments), and MHC-II I-A/I-E (**D** $n_{[WT]} = 17$ for WT, $n_{[IFNγRKO]} = 10$ from four independent experiments) expression on mCherry$^+$ CD45$^-$ cells were analysed by flow cytometry. **E–I** WT and IFNγRKO (KO) tumours expressing mCherry-OVA or ZsGreen-OVA transgenes were admixed 1:1 prior to engraftment in WT (**F, G**), IFNγKO (**H**) or CD8αKO mice (**I**) mice. **E** Experimental design. Graphics created with BioRender. **F** Tumour volumes of WT, IFNγRKO, or admixed tumours taken at endpoint on days 12–14 post-engraftment from three independent experiments for WT:WT (red; $n = 11$) and KO:KO (blue; $n = 10$), and admixed WT:KO tumours (purple; $n = 37$). **G–I** Tumours were harvested and analysed by flow cytometry. Outgrowth of KO tumour cells relative to WT, expressed as percent selection of mCherry$^+$ cells in control WT:WT (red; $n = 6$ (**G, H**), 7 (**I**)), KO:KO (blue; $n = 7$ (**G, H**), 8 (**I**)), or WT:KO (purple; $n = 12$ (**G**), 15 (**H**), 14 (**I**)) tumours at days 14–17 in WT (**G**), IFNγ KO (**H**) and CD8αKO (**I**) mice. Cells were gated on live CD45$^-$ mCherry$^+$. **J, K** WT and IFNγRKO tumours expressing mCherry-OVA or ZsGreen-OVA transgenes were admixed 1:1 in vitro and treated with 10 ng/ml IFNγ when indicated. Ki67 staining ($n = 16$) (**J**) and the ratio between ZsGreen and mCherry (KO/WT) ($n_{[Ctrl]} = 3$; $n_{[IFNγ]} = 17$) (**K**) were assessed by flow cytometry after 2 days. Data are from three independent experiments, each $n$ corresponds to a well (sample). **L** WT and IFNγRKO tumours expressing mCherry-OVA or ZsGreen-OVA transgenes were admixed 1:1 in vitro and treated with 10 ng/ml IFNγ for 10 h. Activated OTI T cells were added to tumour cells for 5 h (OTI:tumours = 2:1). The percentage of apoptotic cells was quantified by flow cytometry using Annexin V staining ($n = 17$). Data are from two independent experiments. **M** WT and IFNγRKO tumours expressing mCherry-OVA or ZsGreen-OVA transgenes were admixed 1:1 prior to engraftment in WT or CD8αKO mice. Tumour volumes of admixed tumours from WT ($n_{[WT:WT]} = 9$, $n_{[KO:KO]} = 7$, $n_{[admix]} = 13$) or CD8αKO ($n_{[WT:WT]} = 7$, $n_{[KO:KO]} = 8$, $n_{[admix]} = 14$) mice was measured on day 12/13 post-engraftment. Data are from two independent experiments (**N**) WT and IFNγRKO (KO) tumours expressing mCherry-OVA or ZsGreen-OVA transgenes were admixed 1:1 (WT:WT = red; KO:KO = blue, WT:KO = purple) prior to engraftment in WT mice and harvested at days 12–14 post-engraftment. Infiltration of lymphocyte populations as a percent of total CD45$^+$ cells from admixed tumours in WT mice ($n_{[WT:WT]} = 11$, $n_{[KO:KO]} = 10$, $n_{[admix]} = 18$) was analysed by flow cytometry. Each $n$ is a tumour. Data are pooled from two or more independent experiments unless otherwise indicated. All data show mean ± SEM with $p$ values by non-parametric two-sided Mann–Whitney $t$ tests for comparisons between two groups, two-sided paired $t$ tests for paired values, Kruskal–Wallis tests between three groups with multiple comparisons correction using Dunn's method, and two-way ANOVA using Šídák's test for multiple comparisons between multiple two or more groups of data. **A, E** Created in BioRender. Gerard (2024) https://BioRender.com/k40f729.

$Ccr7$), and plasmacytoid DCs ($Siglech$, $Ccr9$). Other myeloid clusters comprised monocytes ($Ly6c2$, $Ifitm6$, $Vcan$) and macrophages ($C1qa$, $Spp1$). Six lymphoid populations were present in this dataset, which were comprised of NK cells ($Ncr1$, $Klrb1c$), CD8$^+$ T cells ($Cd3$, $Cd8a$, $Cd8b1$), regulatory T cells ($Cd4$, $Foxp3$), cycling T cells ($Mki67$, $Cd8a$), stem-like T cells ($Cd8a$, $Tcf7$) and B cells ($Cd19$) (Fig. 3A and Supplementary Fig. 3A). In comparing the relative frequency of CD45$^+$ populations, lymphoid populations were modestly variable between WT and IFNγRKO tumours, whereas the macrophage cluster was expanded in WT compared to IFNγRKO tumours (Fig. 3B). Gene set enrichment analysis (GSEA) of the entire scRNAseq dataset unexpectedly revealed IFNγ-signalling and related pathways such as antigen presentation as top hits from both hallmark and Reactome databases of immune cells from IFNγRKO tumours (Fig. 3C, D), suggesting that a highly inflammatory environment emerges following IFNγR deletion in tumour cells. Using ELISA and Legendplex, we confirmed that IFNγRKO tumours contained significantly higher levels of IFNγ and IL-6, whilst other cytokines such as IFNα, TNFα, M-CSF, IL-4, and IL-10 were similar in concentration (Fig. 3E–H and Supplementary Fig. 3B–D), showing that the increase in inflammatory cytokine milieu detected by sequencing in IFNγRKO tumours was the result of an increase in specific inflammatory cytokines. To investigate whether tumours themselves might drive changes in the microenvironment, we treated WT B16-OVA with IFNγ for the indicated period and assessed the expression of a panel of cytokines, for which only CXCL10 was found to be induced by IFNγ (Supplementary Fig. 3E). The concentration of CXCL10 in supernatants from IFNγRKO tumours was slightly lower than in WT tumours (Supplementary Fig. 3F). The receptor of CXCL10, CXCR3, is predominantly expressed in lymphocytes (Supplementary Fig. 3G). Given that the recruitment of lymphoid cells still occurs in IFNγRKO tumours, the slight decrease in CXCL10 is unlikely to explain differences in microenvironment and milieu observed between WT and IFNγRKO tumours. However, in an admix tumour, the lack of CXCL10 expression by IFNγRKO tumours might contribute to T cells preferentially targeting and killing WT tumours. Given that we did not detect major differences in the secretion of cytokine/chemokine between tumour types, we hypothesised that immune cells were driving differences in the tumour milieu, and employed CellChat on the immune scRNAseq dataset as a tool for dissecting soluble signals and cell–cell communications occurring in tumours. We found CD8$^+$ T cells from IFNγRKO tumours to be main signal receivers from monocytes, whereas mono-macs/

macrophages from WT tumours were primary signal senders in WT tumours (Fig. 3I and Supplementary Fig. 3H). Furthermore, quantifying the differential strength of interactions within soluble signalling pathways showed stronger overall interactions between macrophages and other immune subsets such as Tregs and CD8$^+$ T cells in WT tumours compared to IFNγRKO (Fig. 3J).

Overall, our data indicate that IFNγR deletion in tumour cells triggers a remodelling of the immune response and mediators centred around monocytes/macrophages. This led to our hypothesis that tumour-infiltrating monocytes and tumour-associated macrophages (TAMs) may be key in modulating lymphocyte function in each tumour microenvironment.

**Inflammatory myeloid subsets are enhanced by the tumour microenvironment of IFNγ-insensitive tumours**
Following the observation that differences in cytokine signalling and CD45 subsets were likely to lie within the myeloid population, we subset monocytes and macrophages and identified six clusters with unique gene signatures (Fig. 4A and Supplementary Fig. 4A). In line with recent scRNAseq studies describing murine tumour-infiltrating myeloid cells populations[32–34], we identified three monocytic populations, namely, non-classical monocytes (Cluster 6; $Nr4a1$, $Ifitm6$), tumour-infiltrating monocytes (Cluster 2; $Ly6c2$, $Vcan$), and transitionary mono-mac (Cluster 3; $Adgre1$, $Folr2$). We also described four TAM populations, namely, IFN-stimulated TAMs (Cluster 1; $Nos2$, $Sod2$), regulatory TAMs (Cluster 0; $Arg1$, $Spp1$), angiogenic TAMs (Cluster 5; $Vegfa$), and complement TAMs (Cluster 4; $C1qa$) (Fig. 4B). IFNγRKO tumours were dominated by non-classical and tumour-infiltrating monocyte clusters, and IFN-stimulated TAMs, whereas regulatory TAMs were unique to WT tumours (Fig. 4C). Trajectory analysis supported the branching of pre-macrophage (Cluster 3) into more differentiated macrophage phenotypes, i.e. regulatory TAMs (Cluster 0), IFN-stimulated TAMs (Cluster 1), and angiogenic TAMs (Cluster 5) (Fig. 4D). This suggests that the macrophages present in our model might partially be derived from monocytes. Those monocyte-derived macrophages are known to accumulate over time during tumour progression, and have been suggested to shape immune responses[35,36]. However, the type of macrophages elicited in WT versus IFNγRKO tumours differs. Indeed, module scoring of angiogenic and regulatory TAM signatures confirmed the overall increased presence of these subsets in WT tumours compared to IFNγRKO (Fig. 4E and Supplementary Data 1). Angiogenic and regulatory TAMs are known to be pro-

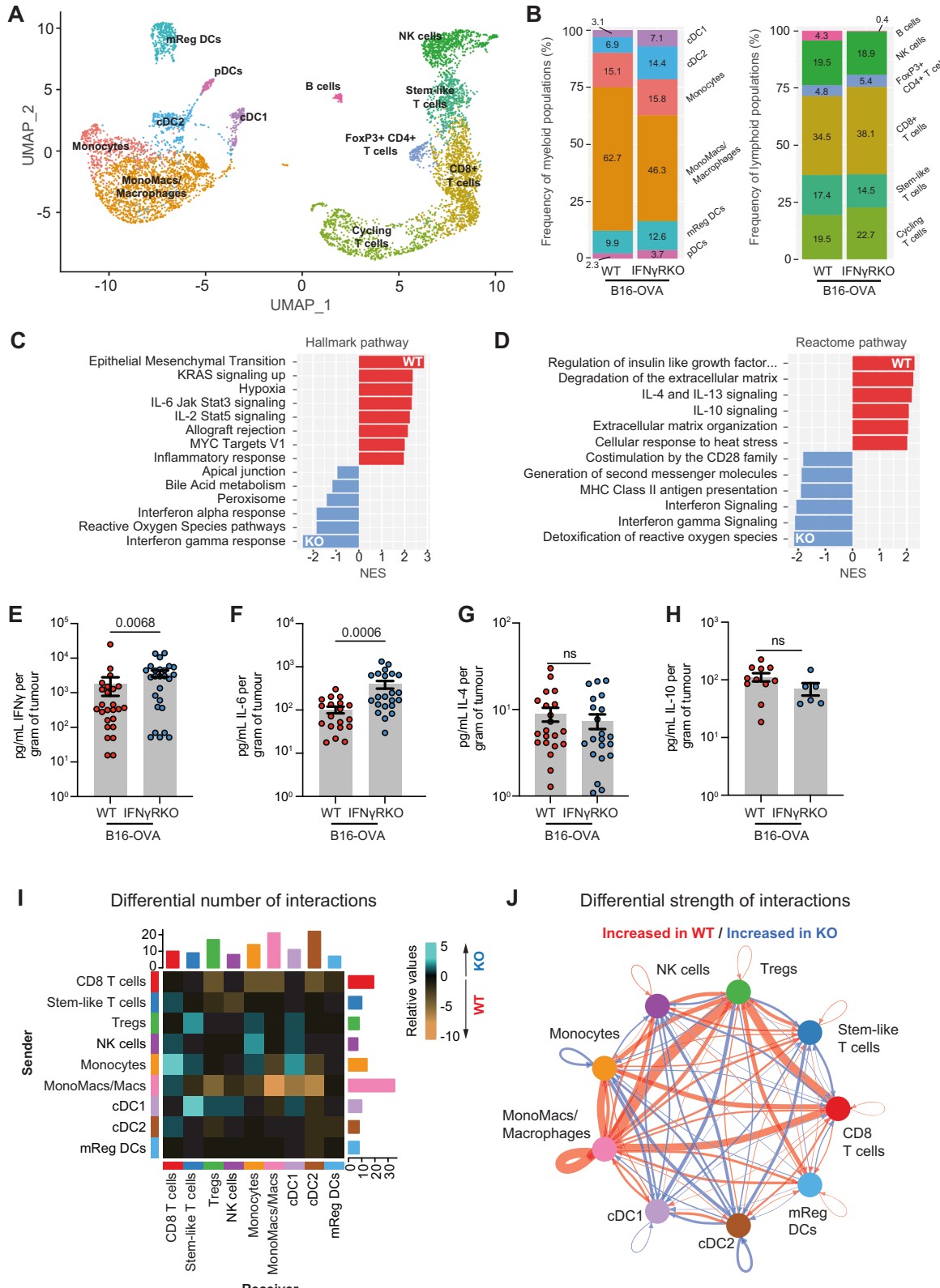

tumourigenic[37,38], and we hypothesise that their absence in IFNγRKO tumours might contribute to tumour control.

We then sought to confirm our scRNAseq findings using flow cytometry and classified monocytes, mono-macs, and macrophages according to Ly6C and MHC class II expression, with gated populations corresponding well to co-expression of macrophage markers such as F4/80, CD204, CD206, and TREM2 (Fig. 4F and Supplementary

Fig. 4B, C). Using this gating strategy to delineate monocytes, mono-macs and macrophages, IFNγRKO and admixed tumours retained a significant proportion of myeloid cells which were monocytic in origin compared to WT (Fig. 4G). We then used spectral flow cytometry for deeper phenotyping of the myeloid landscape (Supplementary Table 2). Unsupervised clustering of spectral flow cytometry data of the CD11b+ CD45+ population revealed an increase in Clusters 1 and 3 in

**Fig. 3 | Single-cell RNAseq analysis of CD45+ tumour-infiltrating cells reveals the presence of enhanced inflammatory milieu in IFNγRKO tumours. A, B** UMAP projection of CD45+ cells and relative abundance of distinct immune populations in WT and IFNγRKO tumours. Clusters show a combined 7014 cells, with 3004 cells from WT tumours, and 4010 cells from IFNγRKO tumours. Gene set enrichment analysis using hallmark (**C**) or Reactome (**D**) databases for identification of enriched signalling pathways, expressed as normalised enrichment scores (NES). Intra-tumoural concentrations of IFNγ ($n_{[WT]}$ = 25, $n_{[KO]}$ = 26) (**E**), IL-6 ($n_{[WT]}$ = 17, $n_{[KO]}$ = 22) (**F**), IL-4 ($n_{[WT]}$ = 21, $n_{[KO]}$ = 22) (**G**), IL-10 ($n_{[WT]}$ = 11, $n_{[KO]}$ = 6) (**H**) measured by ELISA or LegendPlex using supernatants of ex vivo WT (red) and IFNγRKO (blue) dissociated tumours, normalised to tumour weight. Data are pooled from four or more independent experiments. Data show mean ± SEM with $p$ values by non-parametric two-sided Mann–Whitney $t$ tests. **I, J** CellChat ligand–receptor inference analysis was performed on scRNAseq data from (**A**). **I** Heatmap of the differential number of interactions between sender ($y$-axis) and receiver ($x$-axis) populations. Bar plots on each axis represent the sum of all interactions in absolute values for each sender or receiver cell type. **J** Circle plot visualising strength of signalling interactions between immune populations from WT and IFNγRKO tumours. Vertices represent independent populations, and arrows indicate the direction of signals sent, where broader lines represent increased communication probability of signalling interactions.

IFNγRKO tumours, which represent a population of monocytes characterised by Ly6C^hi CD86+ or CD62L+ (Supplementary Fig. 4D–F). In addition, Cluster 7 was increased in WT tumours and corresponded to F4/80+ and CD68+ population, indicative of a more differentiated macrophage phenotype.

We concluded that IFNγR deletion in tumour cells induces a remodelling of the myeloid compartments, with an increase in monocytes and inflammatory macrophages and a concomitant diminution of regulatory TAM.

## Monocyte recruitment is required for controlling IFNγ-insensitive tumours

The CCL2-CCR2 signalling axis plays an indispensable role in the recruitment and trafficking of myeloid populations during infection and inflammation[39,40]. Given that Ly6C^hi inflammatory monocytes primarily depend on CCR2 for tumour infiltration[41] and TAMs originate from recruited monocytes as well as tissue-resident macrophages[36], we engrafted B16-OVA WT or IFNγRKO tumours into CCR2KO mice to determine whether impeding CCR2-dependent recruitment would impact tumour growth. Monocyte recruitment was almost entirely impeded in CCR2KO tumours, which also significantly lacked mono-mac and macrophage populations, indicating that monocytes in this model were indispensable for macrophage recruitment and/or differentiation (Fig. 5A). As found in previous tumour studies using CCR2KO mice[42], monocyte and macrophage populations are replaced by granulocytes, which we observed as a neutrophilic influx especially in IFNγRKO tumours. Importantly, IFNγRKO tumours grew significantly faster than WT when engrafted into CCR2KO mice (Fig. 5B and Supplementary Fig. 5A), suggesting monocyte recruitment is required for the control of IFNγRKO tumours. Neutrophil depletion in CCR2KO mice engrafted with IFNγRKO tumours had no effect on tumour growth (Supplementary Fig. 5B, C), suggesting that the lack of monocyte recruitment, rather than the increase in neutrophils, was responsible for enhanced IFNγRKO tumour growth in CCR2KO mice. Because IFNγRKO tumours are characterised by an increase in monocyte-derived NOS2+ macrophages (Fig. 4A–C), we hypothesised that macrophages controlled IFNγRKO tumour growth through NOS2-induced nitric oxide (NO), which can exert anti-tumour effects[43,44]. To test this, we treated mice engrafted with WT or IFNγRKO tumours with the NOS2 (INOS) inhibitor L-NAME. INOS inhibition increased tumour growth of IFNγRKO tumours (Fig. 5C and Supplementary Fig. 5D). INOS-induced tumour control was less effective in WT tumours (Fig. 5C and Supplementary Fig. 5D), consistent with the fact that fewer NOS2+ macrophages are present in WT compared to IFNγRKO tumours (Fig. 4A–C). Overall, we concluded that monocyte-derived macrophages control growth of IFNγRKO tumours, in part through INOS. In addition, inhibition of monocyte recruitment, through CCR2 deletion, resulted in CD8+ T cell and NK cell expansion in WT, but not IFNγRKO, tumours (Fig. 5D), suggesting that loss of phenotypically immunosuppressive macrophages in WT tumours (Fig. 4A–C) aids in unleashing lymphocyte activity, which in turn improves WT tumour control[45,46]. Loss of CCR2 also decreased the frequency of OVA-specific T cells in IFNγRKO-tumour bearing mice compared with WT mice

(Fig. 5E), which indicates that monocyte recruitment is necessary for the recruitment and/or retention of antigen/tumour-specific CD8+ T cells in IFNγ-insensitive tumours.

Consistent with the recruitment of monocytes from the periphery, imaging of WT and IFNγRKO tumours revealed that Ly6C+ cells resided in or near CD31+ blood vessels, which were found either within the core of the tumour, or at the margin, surrounding the tumour (Fig. 5F and Supplementary Fig. 5E, F). Interestingly, Ly6C+ cells located at the margin also expressed F4/80, suggesting that myeloid cells become excluded from the tumour core as they differentiate (Fig. 5G and Supplementary Fig. 5F). WT and IFNγRKO tumours display gross similar location of the myeloid subsets, showing that the spatial distribution of myeloid cells was not impacted by IFNγR deletion in tumour cells. Focusing on the core of the tumour to specifically investigate the relationship between monocytes and blood vessels, we observed an increased presence of monocytes in contact with blood vessels in IFNγRKO compared to WT tumours, in agreement with enhanced recruitment of monocytes in IFNγRKO tumours (Fig. 5H).

Overall, our results indicate that monocyte infiltration can promote myeloid-CD8+ T cell crosstalk which is important for controlling IFNγ-insensitive tumours.

## IFNγ-insensitive tumours support a pro-inflammatory microenvironment driven by CD8+ T cells which promotes monocyte infiltration and mono-mac differentiation

Our scRNAseq data analysis indicated an increase in the inflammatory milieu following IFNγR deletion. Because the tumours themselves were no longer sensitive to IFNγ, we reasoned that IFNγ was acting on other cells. To explore this, we scored IFNγ signalling in our transcriptomics dataset using a well-established IFNγ gene signature[12]. IFNγR deletion in tumours led to an increase in IFNγ signalling primarily in the myeloid compartment (Fig. 6A), suggesting that IFNγ was important for the remodelling of the myeloid compartment in IFNγRKO tumours. Indeed, the increase in monocyte and mono-macs observed in IFNγRKO tumours (Fig. 4G) was lost if those tumours were engrafted in IFNγKO mice (Fig. 6B). In addition, both WT and IFNγRKO tumours were no longer controlled when engrafted in IFNγKO mice (Fig. 6C and Supplementary Fig. 6A), demonstrating that IFNγ plays an equally important role for both tumour microenvironments. Given the importance of IFNγ for controlling IFNγRKO tumours and remodelling of the myeloid landscape, we sought to identify the cellular source of IFNγ in WT and IFNγRKO tumours. To do so, we used GREAT mice, which carry a reporter for IFNγ whereby the IFNγ promoter controls EYFP expression[47]. Both CD8+ T cells and NK cells were found to be the primary producers of IFNγ in both WT and IFNγRKO tumours during earlier stages of tumour development (i.e. days 7–8 post-engraftment) (Fig. 6D, E). The few CD4+ T cells recruited to tumours were mainly Tregs (Fig. 3A, B) and as such, they did not substantially contribute to IFNγ production (Fig. 6F). IFNγ production peaked ~10 days post-tumour induction (Supplementary Fig. 6B) and was overtaken by CD8+ T cells during latter stages of tumour progression (i.e. days 14–16), regardless of the tumour type. Thus, NK cells lose their capacity to

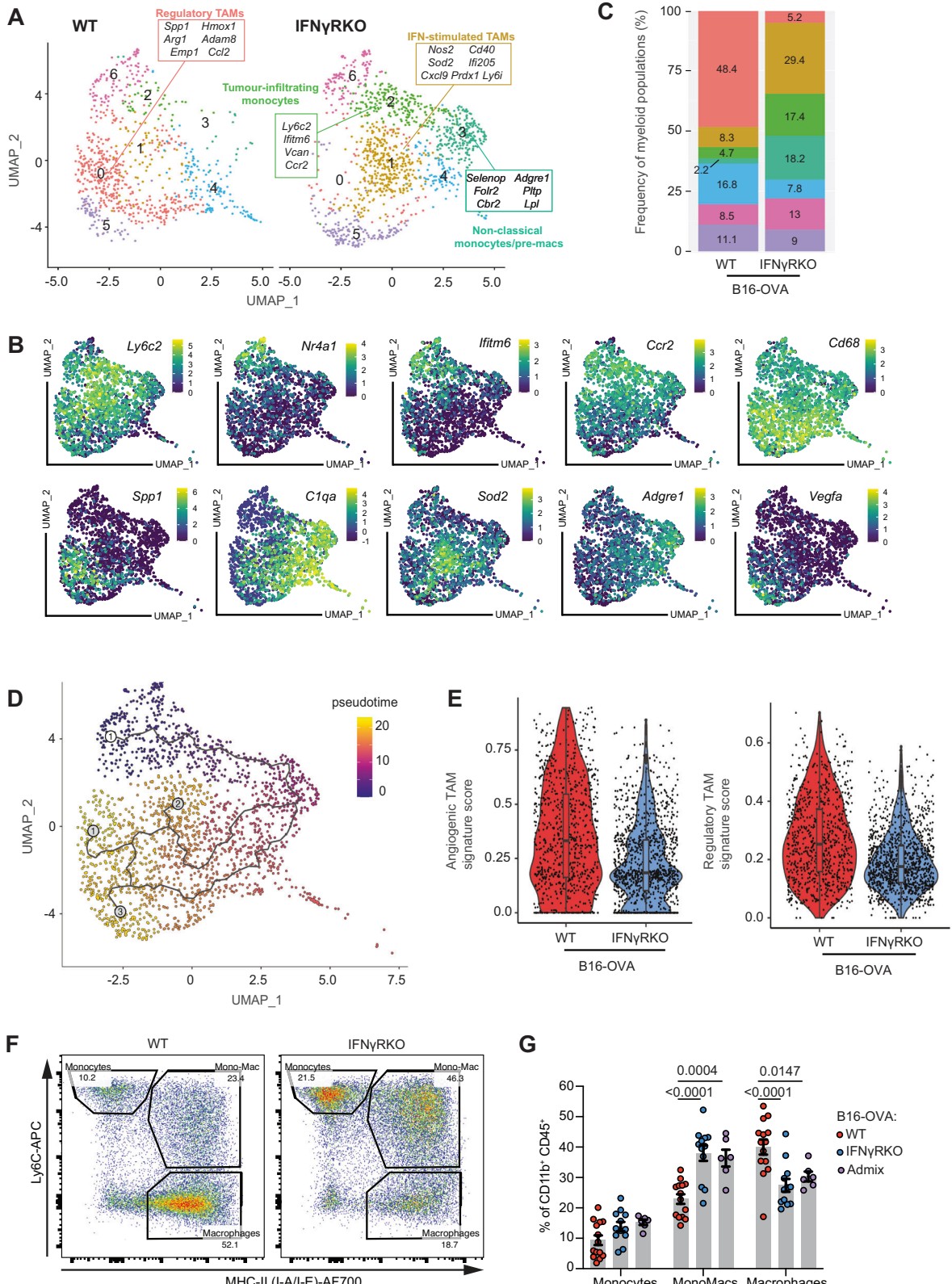

produce IFNγ over time and as such, CD8[+] T cells remain the main source of IFNγ, even in IFNγRKO tumours.

As CD8[+] T cells were important for limiting IFNγRKO tumour growth (Fig. 2M), and the production of IFNγ enables the inflammatory myeloid landscape responsible for IFNγRKO tumour control, we hypothesised that CD8[+] T cells regulate the myeloid landscape in IFNγ-insensitive tumours. As for IFNγKO mice, we did not observe an

increase in monocytes and mono-macs in IFNγRKO compared to WT tumours when engrafted in CD8αKO mice (Supplementary Fig. 6C), indicating that CD8[+] T cells were important to recruit monocytes to IFNγRKO tumours. Similar data were obtained when we specifically depleted CD8[+] T cells by treating mice with a CD8ß depleting antibody before implanting WT or IFNγRKO tumours (Fig. 6G). Deeper pheno-typing revealed that loss of CD8[+] T cells resulted in depletion of

**Fig. 4 | Less differentiated monocyte–macrophage subsets with pro-inflammatory signatures are prominent in IFNγRKO tumours. A–E** Myeloid cells from the scRNAseq data from Fig. 3A were subset and re-clustered. **A** UMAP of WT and IFNγRKO monocyte/macrophage subclusters with distinct myeloid subtypes highlighted by representative gene signatures. **B** Feature plots showing relative gene expression of key monocyte/macrophage genes. **C** Relative frequencies of each subcluster expressed as stacked bar plots. **D** Trajectory analysis overlaid on the UMAP projection of monocyte/macrophage cell clusters, coloured by pseudotime. **E** Violin plots comparing module scoring of angiogenic and regulatory TAMs gene signatures of macrophage subclusters of WT and IFNγRKO tumour samples. Box plots indicate median (middle line), 25th, 75th percentile (box) ($n_{[WT]} = 827$ $n_{[KO]} = 1210$ cells, 3 mice pooled). **F, G** WT, IFNγRKO or admix tumours were engrafted in WT mice and analysed by flow cytometry after 14 days. **F** Representative flow plots of tumour-infiltrating CD45⁺ CD11b⁺ cells from WT and IFNγRKO tumours, gated by Ly6C and MHC-II expression for delineation of monocyte, mono-mac and macrophage populations. Plots are representative of three or more independent experiments. **G** Relative frequencies of each gated population from WT ($n = 14$), IFNγRKO ($n = 12$) or admixed ($n = 6$) tumours. Data are pooled from three independent experiments. Data show mean ± SEM with $p$ values by two-way ANOVA using Šídák's test for multiple comparisons.

classical monocyte Ly6C^hi CX3CR1⁻ or CD86⁺ subsets in both in WT and IFNγRKO tumours (Fig. 6H), but did not affect F4/80⁺ macrophage populations (Supplementary Fig. 6D–F). Ly6C^hi CX3CR1⁻ monocytes have previously been described for their role in renewing intra-tumoural TAM populations[38]. This overall indicates that CD8⁺ T cells are required for the recruitment of classical monocytes with inflammatory and costimulatory properties.

Overall, our data demonstrate that a CD8/monocyte crosstalk is potentiated in tumours insensitive to IFNγ and underlies their control.

### The crosstalk between CD8⁺ T cells and monocytes occurs around vessels in IFNγ-insensitive tumours

To characterise where the crosstalk between CD8⁺ T cells and monocytes occurred, we performed imaging on WT and IFNγRKO tumours. We observed that most blood vessels, labelled with CD31, were surrounded by CD8⁺ T cells (Fig. 7A and Supplementary Fig. 7A, B). At the margin, Ly6C⁺ monocytes were associated with those structures in both WT and IFNγRKO tumours (Supplementary Fig. 7A, B), but due to the large presence of differentiated, F4/80⁺, myeloid cells at the margin, it was unclear whether this could be associated with active recruitment. We therefore focused on blood vessels present in the core of the tumour for quantification. To infer whether the presence of monocytes in or close to vessels correlated with the presence of CD8⁺ T cells, we focused on vessels that contained Ly6C⁺ cells and quantified the proportion of those structures that included CD8⁺ T cells. More than 80% of vessels that contained Ly6C⁺ cells also contained CD8⁺ T cells, regardless of the tumour type (Fig. 7B), in agreement with the fact that CD8⁺ T cells are required for monocyte recruitment in both WT and IFNγRKO tumours. To understand whether CD8⁺ T cells had the same ability to recruit monocytes in WT and IFNγRKO tumours, we focused on vessels that were in contact with CD8⁺ T cells and quantified the proportion of those structures that contained Ly6C⁺ cells. While almost all CD8/vessel structures contained Ly6C⁺ cells in IFNγRKO tumours, only 60% did so in WT tumours (Fig. 7C), consistent with the CD8⁺ T cell-dependent increase in monocyte recruitment observed in IFNγRKO tumours. Thus, while monocytes need CD8⁺ T cells for their recruitment regardless of the tumour type, inhibition of IFNγ sensing in tumours leads to an increase in the ability of CD8⁺ T cells to recruit monocytes. These findings are consistent with studies which describe the role of pro-inflammatory cytokines such as IFNγ in enabling leukocyte adhesion and transendothelial migration through integrin[48,49] or MHC class II upregulation[50]. In IFNγRKO tumours, higher IFNγ levels may increase adhesion of lymphocytes and monocytes to intra-tumoural endothelium, which we observe as a quantifiable increase in these cell–cell interactions. Consistent with this, IFNγ, highlighted using GREAT mice, was produced by CD8⁺ T cells and occurred primarily around blood vessels and in close proximity to some of the infiltrating monocytes (Fig. 7D, E). This suggested that the strategic positioning of IFNγ-producing CD8⁺ T cells either supported monocyte recruitment, and/or allowed for monocytes to receive differentiation signals as soon as they entered tumours. Although monocytes are not necessarily directly in contact with IFNγ-producing cells, IFNγ spread can reach between 3 and 30

cells depending on models and T cell density[51–53], which is in line with the close proximity between IFNγ-producing cells and monocytes we observed (Fig. 7E).

Overall, our data demonstrate that the CD8/monocyte crosstalk occurs around vessels in IFNγ-insensitive tumours, where CD8 T cells recruit monocytes and skew their differentiation.

### Elevated CD8-monocyte immune signature scoring across multiple human cancer types

Our data using mouse models points to a remodelling of the immune response driven by inhibition of IFNγ receptor or signalling. To explore whether this is also elicited in human tumours, we elected to investigate whether enrichment scoring using a combined CD8-monocyte signature would be increased in TCGA RNAseq datasets in which the patient tumours harboured mutations in *IFNGR1/2*, *JAK1/2*, or *STAT1*. Only datasets in which the calculated variant consequences (i.e. VEP IMPACT score) deemed high or moderate were included for analysis. Using single-sample GSEA, multiple tumour types scored higher for CD8-monocyte enrichment in IFNγ-pathway mutation-containing datasets compared to controls (Fig. 8A). This is consistent with our data in mice, and suggests that mutations in the IFNγ-pathway can drive an enhanced CD8⁺ T cell/monocyte immune response.

Our data suggested that the interplay between CD8⁺ T cells and myeloid cells occurred around blood vessels. To explore this in human tumours, we performed analysis of publicly available spatial transcriptomic datasets from the 10X Genomics Visium platform to investigate IFNγ response and CD8-monocyte gene signatures on human lung and colon cancer samples. Across both tumour types, CD8-monocyte gene signature expression coincided spatially with hallmark IFNγ response signatures and endothelial cell markers (Fig. 8B, C and Supplementary Fig. 8A, B). These regions were also correlated with M2 macrophage signatures, indicating that IFNγ-response regions are likely associated with active immune infiltration rather than specific immune subsets due to limits in spatial resolution of individual cell types (Supplementary Fig. 8C, D). Other hallmark pathways such as TGFβ-signalling or hypoxia showed incongruent overlap with immune-rich regions, providing evidence that regions of opposing immune function or activity are likely to be compartmentalised.

Overall, analysis of human datasets corresponds to our murine model and shows that CD8-monocyte signatures can be found in tumours with mutations in the IFNγ-pathway. Human spatial datasets also mirror the association of CD8⁺ T cells and monocytes in our imaging studies, highlighting the importance of their co-operativity in the tumour microenvironment.

## Discussion

In this study, we demonstrate a novel mechanism in which IFNγ-insensitive tumours trigger remodelling of the tumour microenvironment through accumulation of IFNγ which leads to effective tumour control. We show that IFNγRKO tumours were dominated by inflammatory monocytic and pre-macrophage subsets compared to archetypal TAMs in WT tumours, and this mechanism relied on CCR2-dependent myeloid recruitment. Moreover, we uncovered a monocyte-CD8⁺ T cell reciprocity where depletion of either monocytes

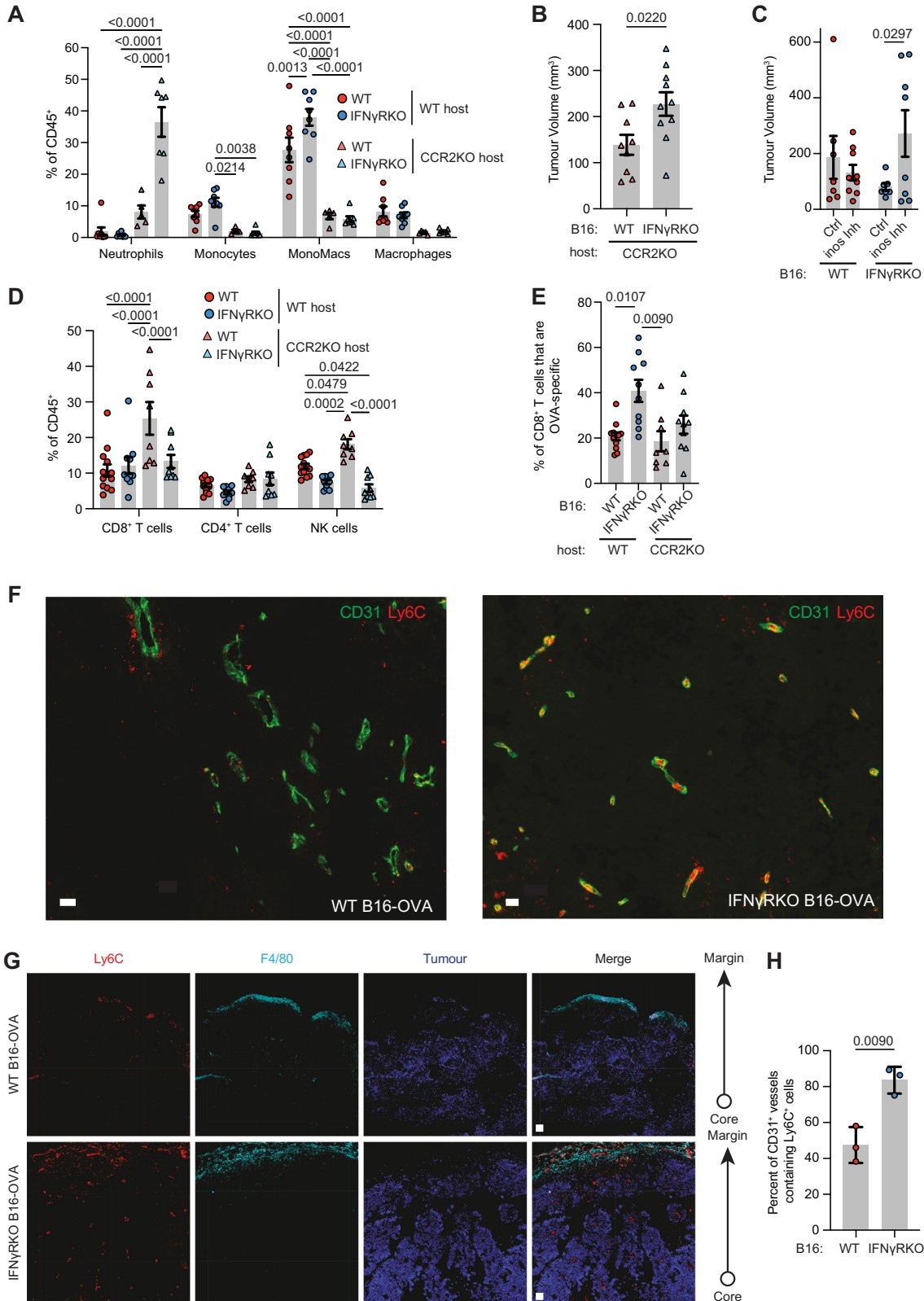

or CD8+ T cells impaired control of IFNγRKO tumours; loss of monocyte infiltration impeded infiltration of tumour-specific T cells, and CD8+ T cell depletion resulted in loss of inflammatory monocyte subsets. The phenomenon of immune sensitisation following IFNγ-signalling ablation is not restricted to B16F10 melanoma, as other groups have reported similar findings in mammary, colon, pancreatic and lung tumour models in both Balb/c and C57Bl/6 animals[23,54,55].

Incidentally, our results provided a mechanistic basis for reports which showed that IFNγ-pathway mutations undergo positive selection during in vivo CRISPR screens[54], and perhaps why mutations in the IFNγ pathway resulted in sensitisation towards ICB responses[23].

We demonstrate that low, baseline levels of MHC class I can be sufficient for eliciting strong CD8+ T cell-dependent anti-tumour activity. Consistent with this, meta-analysis studies failed to

**Fig. 5 | Control of IFNγRKO tumours is diminished in CCR2KO mice following lack of monocyte recruitment. A**, **B** WT (red) or IFNγRKO (blue) tumours were engrafted in WT (circle) or CCR2KO (triangle) mice. **A** Infiltration of myeloid populations relative to total CD45$^+$ cells in WT or IFNγRKO tumours engrafted into WT ($n_{[WT\ tumours]} = 8$, $n_{[KO\ tumours]} = 8$) or CCR2KO ($n_{[WT\ tumours]} = 5$, $n_{[KO\ tumours]} = 7$) mice. **B** Tumour volumes of WT ($n = 9$) or IFNγRKO ($n = 10$) tumours in CCR2KO hosts measured at endpoint on day 13 post-engraftment. Data are from two independent experiments. **C** WT (red) or IFNγRKO (blue) tumours were engrafted in WT mice. Mice were treated with the INOS inhibitor L-NAME (INOS Inh) when indicated. Tumour volumes were measured on day 13 post-engraftment ($n_{[Ctrl]} = 7$, $n_{[L-NAME]} = 8$). **D**, **E** WT (red) or IFNγRKO (blue) tumours were engrafted in WT (circle; $n_{[WT\ tumours]} = 13$, $n_{[KO\ tumours]} = 10$) or CCR2KO (triangle; $n_{[WT\ tumours]} = 8$,

$n_{[KO\ tumours]} = 9$) mice. **D** Infiltration of T cells and NK cells relative to total CD45$^+$ cells. **E** Frequency of OVA-specific T cells as a proportion of CD8$^+$ T cells in WT ($n_{[WT\ tumours]} = 12$, $n_{[KO\ tumours]} = 10$) and CCR2KO ($n_{[WT\ tumours]} = 8$, $n_{[KO\ tumours]} = 10$) mice. Data are from two independent experiments. **F**–**H** Frozen sections from WT or IFNγRKO tumours engrafted in WT mice were stained with the indicated markers and imaged. Representative images indicating location of Ly6C (red) and F4/80 (Cyan) expressing cells relative to CD31$^+$ (green) blood vessels. Scale bar = 40 μm (**F**), and 100 μm (**G**). **H** Graph shows the percentage of blood vessels that are in contact with Ly6C$^+$ cells. Each dot is a tumour ($n = 3$). All data show mean ± SEM with $p$ values by non-parametric two-sided Mann–Whitney $t$ tests for comparisons between two groups, and two-way ANOVA using Šídák's test for multiple comparisons between multiple two or more groups of data.

demonstrate a strong association between stable/progressive disease and loss of HLA[56]. In addition, MHC-I negative tumours only indicate poor survival when PD-L1 is concomitantly expressed, and tumours which are negative for both show no difference in survival[57]. Finally, patient-derived melanoma cell lines carrying JAK1 or JAK2 knockouts retained basal MHC-I expression and the capacity to activate tumour-specific T cells in vitro[57]. In this context, our study strongly suggests that IFNγ likely plays a significant role in stabilising antigen presentation by myeloid subsets, especially during early tumourigenesis, which in turn modulates long-term tumour-specific T cell persistence. We and others have recently highlighted the importance of such myeloid-T cell interactions[45] and other studies have shown that macrophages are capable of cross-presenting tumour antigens to CD8$^+$ T cells[58,59].

Although it is often assumed that the most important function of CD8$^+$ T cells is direct killing of tumour cells, our data highlight that their role in promoting a cytokine environment permissive for tumour control is equally as important. CD8$^+$ T cell-derived IFNγ is known for mobilising rapid effector functions of innate populations during secondary recall responses during infection, and myeloid cells lacking IFNγR expression fail to control pathogens[60]. We found that IFNγ production by CD8$^+$ T cells mostly occurred around blood vessels in tumours. Perivascular immune niches that contain CD8$^+$ T cells, DCs and activated macrophages have been correlated with anti-tumour immunity[61], and likely supports IFNγ production in this strategic perivascular area, inducing rapid differentiation of monocytes into macrophages. Of note, IFNγ-mediated macrophage differentiation can also be carried by CD4$^+$ T cells in other tumour models[62] and might therefore not always be solely dependent on CD8$^+$ T cells. In ICB-sensitive models such as MC38 murine colon adenocarcinoma, anti-PD-L1 drove a significantly more pro-inflammatory macrophage phenotype compared to untreated tumours, and IFNγR$^{-/-}$ bone marrow transferred into tumour-bearing WT mice were only able to produce M2-like macrophages[63]. Accordingly, analysis of patient outcomes following ICB ± chemotherapy[64–67] or adoptive cell therapy[68] demonstrates that the presence of inflammatory monocytes and M1-like macrophages are substantially better in predicting therapy response than traditional biomarkers such as TMB or PD-1 expression[69–71].

One major question remaining is how these plastic and transitionary myeloid populations shift during disease progression, and whether remodelling of the anti-tumour response also occurs early in human cancers. Sampling of early-stage tumours suggested that tumour CD14$^+$ cells were not primarily immunosuppressive against T cell cytokine production or proliferation[72]. Our admixed model suggests that immune-mediated clonal selection occurs over a longer period of time, where WT and IFNγRKO tumour cells no longer face equal pressure by CD8$^+$ T cells, and loss of the ability to kill MHC-I$^{low}$ cells may be attributed to an increasingly immunosuppressive myeloid compartment. Dissecting out the influence of myeloid cells on CD8$^+$ T cell function in a longitudinal manner would assist in answering this fundamental mechanism of escape and selection relative to human

clinical scenarios where tumours are heterogeneous in their mutations.

Finally, a monocyte/CD8 T cell signature has been recently described in multiple studies as a predictor of good clinical response following multiple types of immunotherapies[64,73]. Our data demonstrate that monocyte/CD8 T cell crosstalk indeed enhances tumour control, in particular for tumours that are less immunogenic due to inhibition of the IFNγ pathway and low MHC expression, and importantly, gives a potential mechanism explaining how this crosstalk potentiates anti-tumour responses, beyond IFNγ-insensitive tumours.

## Methods
### Mice
C57BL/6J (B6) WT male mice were purchased from Charles River (JAX number−000664) and housed 1–2 weeks before experimentation. OT-I (JAX stock number: 003831) mice were bred with CD45.1 mice (The Jackson Laboratory−002014) to generate congenically marked OT-I CD45.1 cells. C57BL/6J mice were purchased from Charles River, UK (JAX stock number: 000664). IFNγ-GREAT$^{YFP}$ (JAX number−017580), CD8αKO (JAX number−002665), IFNγKO (JAX number−002287), and CCR2KO (JAX number−004999) were housed and bred under specific pathogen-free/SPF conditions in the in-house animal facilities at the University of Oxford. All mice are from C57BL/6 background. Experimental and control animals were co-housed and kept in individually ventilated cages supplemented with environmental enrichment at 20–24 °C, 45–64% humidity, and 12 h light/dark cycles. Mice were euthanized by CO$_2$ asphyxiation followed by cervical dissociation. Both males and female mice between the age of 6–14-week-old were used. Age- and sex-matched mice of indicated genotypes were randomly allocated to groups for comparison. All experiments involving mice were conducted in agreement with the United Kingdom Animal Scientific Procedures Act of 1986 and performed in accordance with approved experimental procedures by the Home Office and the Local Ethics Reviews Committee (University of Oxford) under UK project licenses P4BEAEBB5 and PP3609558.

### Cell line generation and culture conditions
B16F10 Tyr$^{-/-}$ expressing mCherry and ovalbumin (B16-OVA) was kindly provided by Dr. Edward Roberts from the Beatson Institute (Glasgow, UK). Knockout of murine IFNGR1 using CRISPR-Cas9-mediated gene editing using protocols described by Ran et al.[74]. Briefly, the single guide RNA (sgRNA) sequences targeting exon 2 of murine IFNγR1 were cloned into the pX458 backbone (Addgene) containing Cas9 expression and GFP expression, followed by validation via Sanger sequencing. Target murine cell lines were transiently transfected with pX458 sgIFNγR1 using Lipofectamine 3000 (Invitrogen, Cat. L3000001). After 48 h, cells were single-cell sorted into individual wells of a 96-well plate. Single-cell clones were expanded and stimulated with 1 ng/mL recombinant murine IFNγ (Peprotech, Cat. 315-05). Clones were stained for MHC-I H2D$^b$ expression by flow cytometry analysis. Five individual clones unresponsive to IFNγ

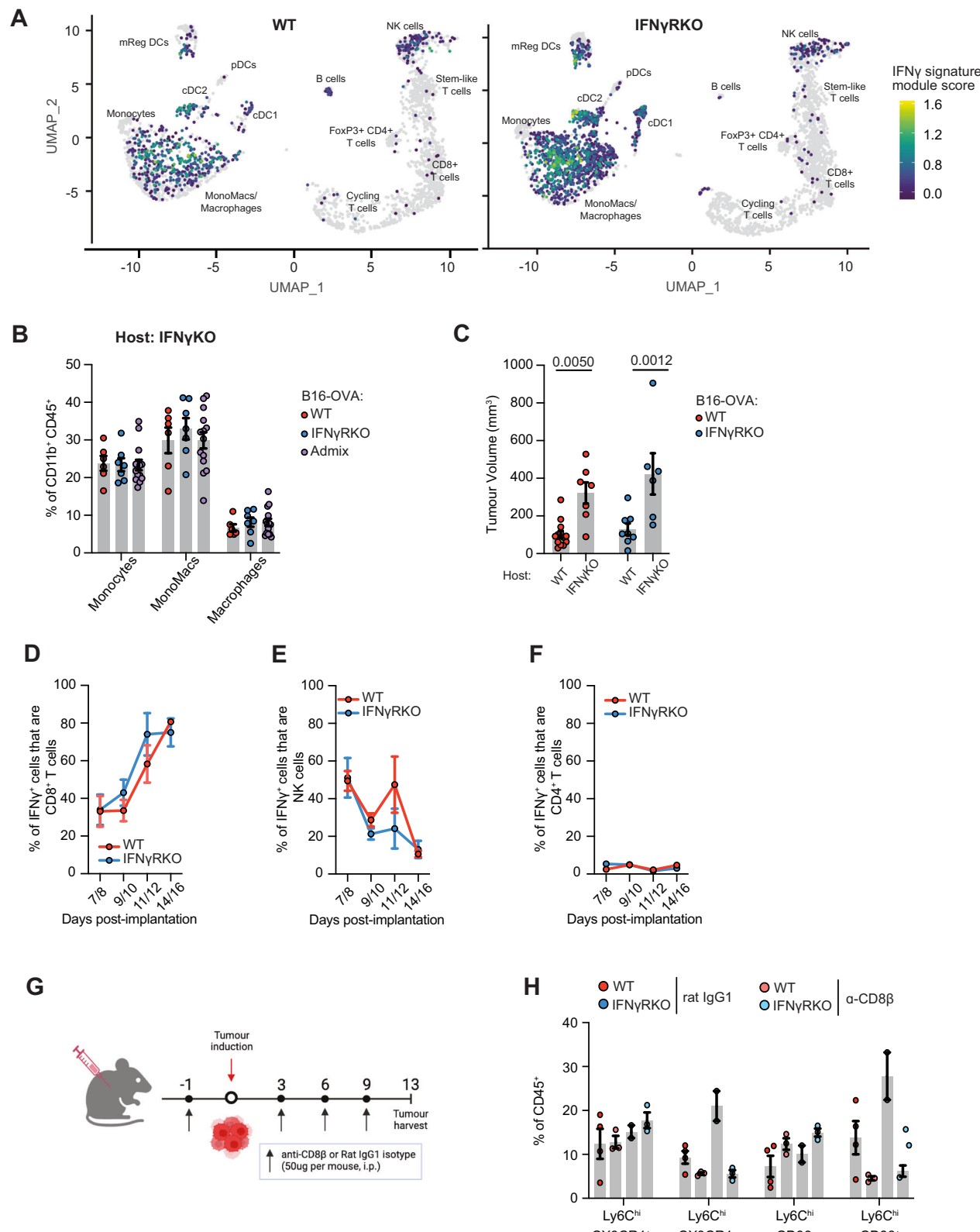

stimulation were pooled to form the final cell line used in subsequent experiments. For the double knockout of H2K[b] and H2D[b], similar strategy was used, targeting exon 2. Clones were stained for MHC-I H2D[b] and H2K[b] expression after IFNγ treatment by flow cytometry. Five individual clones were pooled to form the final cell line used in subsequent experiments. For generation of the cell line expressing IFNγR1 Y445A, wild-type and mutated sequences were synthesised as gBlock

gene fragments (Integrated DNA Technologies, Inc.) and cloned into a pLV lentiviral backbone containing puromycin resistance. Wild-type IFNGR1 and Y445A were re-expressed in the B16 IFNγR1KO cell line and selected for puromycin-resistant cells. Resulting cell lines were validated by stimulation with rmIFNγ at 10 ng/mL for MHC class I re-sensitisation. All cell lines were cultured in RPMI 1640 (Gibco, Cat. 21870-076) supplemented with 10% FCS (Sigma, Cat. F9665-500ML),

**Fig. 6 | Recruitment of monocytes is driven by CD8⁺ T cell-derived IFNγ.**
**A** Module scoring of an IFNγ gene signature on the single-cell dataset from Fig. 3A. **B** Infiltration of myeloid populations relative to total CD11b⁺CD45⁺ cells in WT (red; $n = 6$), IFNγRKO (blue; $n = 7$) or admixed (purple; $n = 14$) tumours engrafted into IFNγKO mice. Data are pooled from two independent experiments. **C** Tumour volumes of WT (red) or IFNγRKO (blue) tumours measured on day 10 post-engraftment of WT ($n_{[WT\ tumours]} = 13$, $n_{[KO\ tumours]} = 8$) or IFNγKO ($n_{[WT\ tumours]} = 7$, $n_{[KO\ tumours]} = 6$) mice. Data are pooled from two independent experiments. **D–F** WT or IFNγRKO tumours were engrafted in GREAT mice and harvested when indicated. Percentage of cells which are IFNγ+, as measured by EYFP expression by tumour-infiltrating CD8⁺ T cells (**D**), NK cells (**E**), and CD4⁺ T cells (**F**). Data are pooled from four independent experiments, with timepoints varying between experiments (WT tumours: $n_{[day7/8]} = 8$, $n_{[day9/10]} = 16$, $n_{[day11/12]} = 8$, $n_{[day14/16]} = 4$; KO tumours: $n_{[day7/8]} = 5$, $n_{[day9/10]} = 13$, $n_{[day11/12]} = 6$, $n_{[day14/16]} = 5$ day). **G, H** Antibody depletion of CD8⁺ T cells using anti-CD8β before and following tumour engraftment. **G** Experimental design. Graphics created with BioRender. **H** Frequency of specific Ly6Cʰⁱ monocyte subsets following CD8⁺ T cell depletion was analysed by flow cytometry 13 days post-engraftment ($n_{[WT-isotype]} = 4$, $n_{[WT-anti-CD8β]} = 3$, $n_{[KO-isotype]} = 2$, $n_{[KO-anti-CD8β]} = 3$). All data show mean ± SEM with Kruskal–Wallis testing between three groups with multiple comparisons correction using Dunn's method, and two-way ANOVA using Šídák's test for multiple comparisons between multiple two or more groups of data. **G** Created in BioRender. Gerard (2024) https://BioRender.com/k40f729.

and 1X penicillin/streptomycin/L-glutamine (Gibco, Cat. 10378-016) (referred to as R10 medium). Cell lines were kept at 37 °C in 5% CO₂ and routinely checked for mycoplasma contamination via LookOut Mycoplasma PCR detection kit (Sigma, Cat. MP0035). Guide sequences are provided in Supplementary Table 1.

## Tumour induction and administration of immune-modifying agents
Cell lines were harvested at 50–70% confluency on the day of tumour injections using trypsin–EDTA (Sigma, Cat. T3924-500ML) and washed twice in PBS prior to resuspension at desired cell concentrations in PBS or PBS + 25% Matrigel (Corning, Cat. 354262, or Cat. 256231). Mice were anaesthetised using isoflurane (Zoetis) and flanks were shaved prior to injection. Tumours were typically engrafted subcutaneously in the right and/or left ventral flanks at a cell concentration of $1.0–3.5 \times 10^6$ cells/mL, resulting in engraftment of $1 \times 10^5$ or $3.5 \times 10^5$ cells per 100 µL injection. Tumours were measured after 5–7 days post-injection using callipers and monitored every other day for humane endpoints continuously until experimental termination. Tumour volumes were calculated using $[(L \times W \times H)/2]$ formula in mm³. Humane endpoint was defined as a tumour length of 12 mm in any direction, as permitted by the AWERB ethics committee. Mice were killed before their tumour reached humane endpoint.

In some experiments, mice were treated with CD8 or NK1.1 depleting antibodies or isotype controls (BioXCell, anti-CD8β Cat. BE0223, and anti-NK1.1 Cat. BE0036). Mice were injected intraperitonially with 50 µg antibodies every 2–3 days and monitored daily. For depleting neutrophils, mice were treated every 2 days with a combination of 25 µg anti-Ly6G (1A8, Biolegend Cat. 127649) and 50 µg anti-rat Kappa immunoglobulin (MAR18.5, Thermo Fisher Cat. I-2026)[75] or corresponding isotype controls from the start. For inhibiting INOS, the INOS inhibitor L-NAME (Merck Cat. N5751) was added to drinking water at 1 mg/ml throughout the experiment.

## Tissue processing
At indicated timepoints, mice were sacrificed and tumours were measured, excised, and weighed before processing. Tumours were dilacerated using scalpels to obtain <1 mm sized pieces and resuspended in R10 supplemented with 1 mg/mL Liberase TL (Roche, Cat. 5401020001) and 10 µg/mL DNase I (Roche, Cat. 11284932001) for enzymatic digestion. Tumour suspensions were incubated at 37 °C for 30 min before physical dissociation of remaining fragments through 70 µm cell strainers to obtain single-cell suspensions.

## ELISAs and LegendPlex assays
Tumour supernatants were collected before enzymatic digestion following mechanical dilaceration into 1 mL of R10 media. Samples were frozen at −20 °C and thawed on ice prior to assaying. IFNγ concentrations were determined using an IFNγ mouse uncoated ELISA kit (Invitrogen, Cat. 88-7314-77) following manufacturer's protocols. Obtained IFNγ concentrations using the standard curve were normalised to tumour weights. For LEGENDplex™ Mouse Cytokine Release Syndrome Panel (13-plex) assays (Biolegend, Cat. 741024), 25 µL of supernatant was used following manufacturer's protocols and samples were analysed by BD LSRFortessa.

## Flow cytometry
Single-cell suspensions from cell lines or tumours were plated into 96-well V bottom plates at concentration of $2.5 \times 10^6$ cells or less. Cells were washed 1X using PBS prior to addition of viability dyes (Biolegend, Zombie NIR, Cat. 423106, or Zombie UV, Cat. 423108) according to manufacturers' instructions. Samples were incubated with anti-CD16/anti-CD32 blocking antibodies (Biolegend, Cat. 101302) for 20 min at 4 °C, followed by fluorochrome-conjugated antibodies against extracellular markers for 30 min at 4 °C. Cells were washed with FACS–EDTA buffer (2% FCS, 2.5 mM EDTA, and 0.01% sodium azide in PBS) and resuspended in 2% paraformaldehyde (Alfa Aesar, Cat. 43368.9M) in PBS for a 20-min fixation at 4 °C. Cells were washed before resuspension in FACS–EDTA buffer and stored until analysis. For experiments where tetramers were used, tetramers were diluted in FACS–EDTA and incubated with samples in-between the Fc- blocking and antibody staining steps. Tetramers were obtained from the NIH Tetramer Core Facility (Atlanta, GA, USA). All flow cytometry samples were recorded using BDFACSDiva (v8.0), and analysed by BD Fortessa X-20, BD LSRII, or Cytek Aurora as indicated per experiment. Flow cytometry data were analysed using FlowJo V.10 (BD), or OMIQ (Dotmatics).

## In vitro proliferation and cytotoxicity assays
B16-OVA WT and IFNγRKO expressing mCherry or ZsGreen, respectively, were admixed in 96-well plates and treated with 1–10 ng/ml IFNγ at 37 °C, 5% CO₂.

For proliferation assay, after 2 days, cells were harvested and proliferation was assessed by flow cytometry using intracellular staining of Ki67 (Biolegend, Cat. 652408). Outgrowth of tumour types was calculated as the ratio between the number of ZsGreen B16-OVA IFNγRKO and mCherry B16-OVA WT. Changing fluorophores did not affect the data.

For cytotoxicity assay, CD8⁺ T cells from OTI mice were isolated from the spleen of 8–10-week-old animals and activated with 50 ng/ml of the OVA peptide SIINFEKL for 5 days in the presence of 20 U/ml IL-2. OTI cells were deposited on admixed B16-OVA cells 1 day after plating (OTI: B16-OVA = 2:1). B16-OVA cell death was assessed with the Annexin V apoptosis detection kit (eBioscience, Cat. 88-8007-72) by flow cytometry.

## Tissue fixation, cryosectioning, and imaging
Whole intact tumours were harvested on days 10–13 post-engraftment and immediately immersed in fixative solution (1% PFA, 75 mM L-lysine [Sigma, Cat. L5501], 10 mM sodium m-periodate [Thermo Scientific Pierce, Cat. 20504], diluted in 0.2 M PBS adjusted to pH 7.4) for 16–20 h at 4 °C under gentle agitation. Fixative solution was discarded,

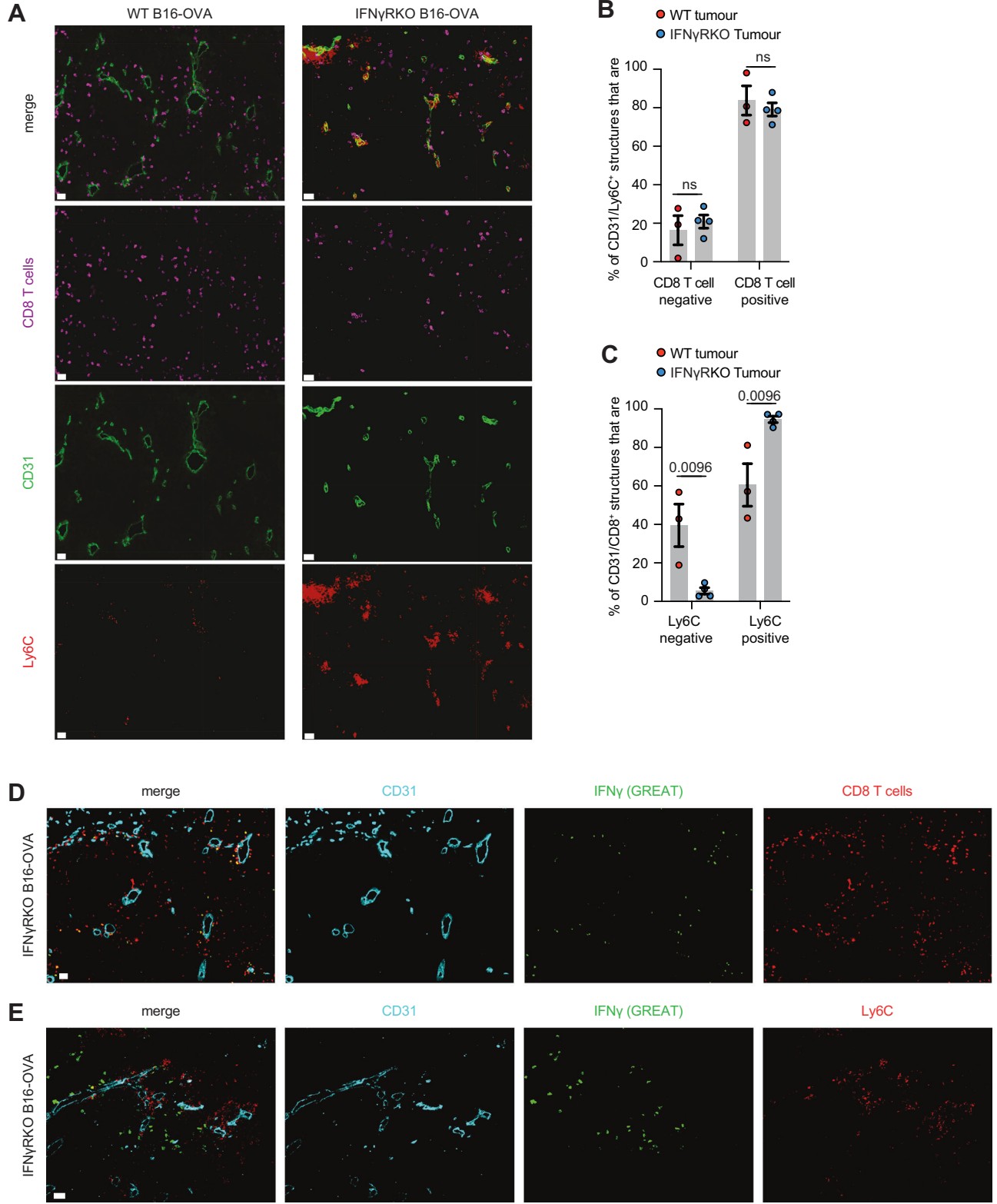

**Fig. 7 | CD8-monocyte crosstalk occurs close to blood vessels. A−C** WT or IFNγRKO tumours were grown in WT mice. Frozen sections from WT or IFNγRKO tumours were stained with the indicated markers and imaged. **A** Representative immunofluorescence images taken at the core of WT and IFNγRKO tumours showing the location of Ly6C (red) and CD8 (magenta) cells relative to CD31+ blood vessels (green). Scale bar = 30 μm. **B** Bar graph shows the percentage of CD31+/ Ly6C+ structures that contain CD8+ T cells Each dot is a tumour (*n* = 3). **C** Bar graph shows the percentage of CD31+/CD8+ T cell structures that contain Ly6C cells. Each dot is a tumour (*n* = 3). **D**, **E** IFNγRKO tumours were grown in GREAT mice. Frozen sections from IFNγRKO tumours were stained with the indicated markers and imaged. **D** Representative immunofluorescence images taken at the core of IFNγRKO tumours showing the location of CD8+ cells (red) and IFNγ expressing cells (green) relative to CD31+ blood vessels (cyan). **E** Representative immuno-fluorescence images taken at the core of IFNγRKO tumours showing the location of Ly6C+ cells (red) and IFNγ expressing cells (green) relative to CD31+ blood vessels (cyan). This is a representative example of three independent tumours. Scale bar = 30 μm All data show mean ± SEM with two-way ANOVA using Šídák's test for multiple comparisons between two or more groups of data.

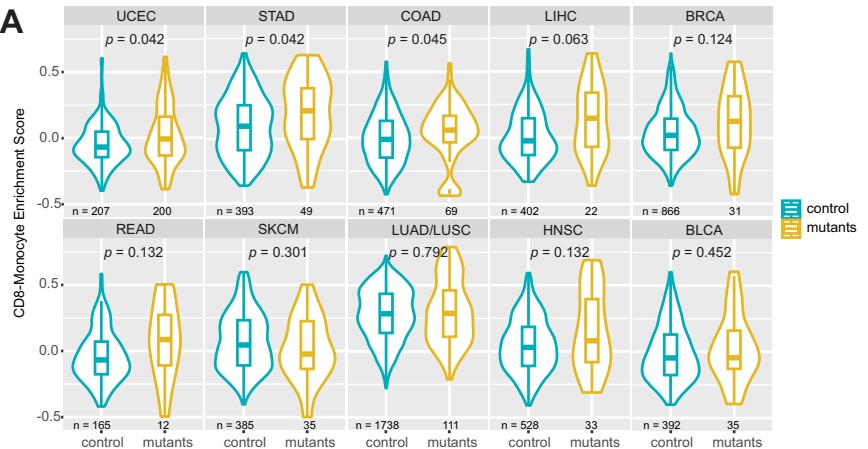

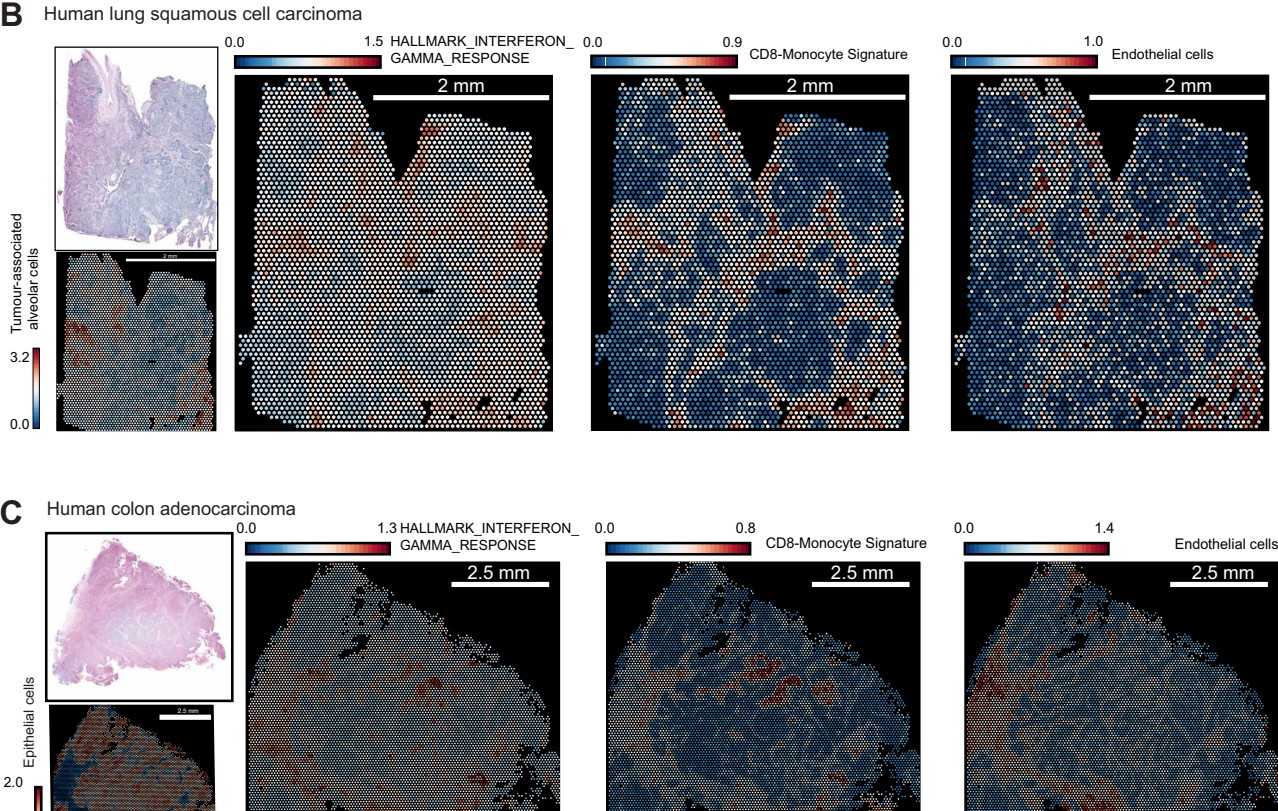

**Fig. 8 | CD8-monocyte signatures are elevated in human tumours with identified IFNγ-pathway mutations, and spatially overlap with IFNγ response signatures. A** Enrichment scoring of CD8-monocyte gene signatures using single gene set enrichment analysis of human TCGA RNAseq datasets with (yellow) and without (blue) IFNγ-pathway mutations. Normalised gene counts from each tumour type were taken from samples which had moderate or high impact *IFNGR1/2, JAK1/2, STAT1* mutations determined by whole-exome sequencing. Box plots indicate median (middle line), 25th, 75th percentile (box). Number of samples included for analysis are indicated for each sample set, and statistical testing using two-sided Wilcoxon signed-rank test with adjust *p* values by false discovery rate testing is shown. **B** Analysis of 10X Genomics Visium datasets for hallmark IFNγ response, CD8-monocyte, and endothelial cell gene signatures for human lung squamous cell carcinoma (**B**) or colon adenocarcinoma (**C**) samples. Gene set expression is indicated by heatmap, where colours represent log-normalised average expression.

and tumours were washed using 1X PBS for 1 h at 4 °C under gentle agitation to remove remaining PFA. Tumours were then resuspended in 30% sucrose (w/v, diluted in pH 7.4 PBS) for 24–36 h at 4 °C without agitation, until tissue was no longer floating. Tumours were cryogenically frozen in OCT compound (Thermo Fisher, Cat. 15212776) using methanol and dry ice bath, and stored at −80 °C until cryosectioning. Frozen tumour blocks were cryosectioned at 10 μm thickness using a Leica CM1900UV and mounted onto glass slides (VWR,

Cat. 631-0108). Cryosections were stored at −80 °C until staining. For staining, sections were washed with PBS to remove OCT compound on the slides, and blocked with solutions containing imaging buffer (2% FCS, 0.1% Triton X-100 (Sigma, Cat. X100), 0.01% sodium azide), FcBlock, and species-specific serum depending on the fluorochrome-conjugated antibodies used in each staining panel. Sections were blocked for a minimum of 3 h at room temperature, before incubation with fluorescently conjugated antibodies diluted in blocking solution

overnight at 4 °C. A final wash was performed twice using imaging buffer before the sections were mounted using Fluoromount G (Southern Biotech, Cat. 0100-01) and glass coverslips were placed on top of the sections. Images were collected using Zeiss Axioscan 7 Slide Scanner or Zeiss LSM 980 confocal microscope, and analysed using Imaris software (Bitplane, V10.0).

## Single-cell RNA sequencing

Single-cell suspensions from three B16-OVA WT tumours were labelled with TotalSeq(TM)-C0301 antibody (Biolegend, Cat. 155861), and three IFNγRKO tumours were labelled with TotalSeq(TM)-C0302 (Biolegend, Cat. 155863). Live cells stained with Zombie NIR and CD45 antibody were sorted based on expression of CD45 using a BD FACSAria™ II. Approximately 10,000 cells per sample were loaded onto the 10x Genomics Chromium Controller (Chip K). Gene expression and feature barcoding libraries were prepared using the 10x Genomics Single Cell 5′ Reagent Kits v2 (Dual Index) following the manufacturer user guide (CG000330 Rev B). The final libraries were diluted to ~10 nM for storage. The 10 nM library was denatured and further diluted prior to loading on the NovaSeq6000 sequencing platform (Illumina, v1.5 chemistry, 28 bp/98 bp paired end for gene expression and feature barcoding).

## Analysis of scRNAseq datasets

Sequence reads were mapped using CellRanger multi (version 6.0.0) and the 10x mouse reference transcriptome (version 2020-A). The R package Seurat v4 (v4.0.6)[76] was used in conjunction with other tools for QC, demultiplexing, filtering, and annotation of the dataset. Briefly, singlets were extracted from the dataset, and counts were log normalised and variable features were scaled. Cells having fewer than 500 or greater than 6000 detected genes were filtered out. Cells in which 5% of the UMIs represent mitochondrial protein-coding genes or more than 20% of large gene contents were also filtered. Lastly, decontX[77] was used to determine contamination of droplets with ambient RNA. The filtered dataset was scaled, log normalised, and variable features were identified using the functions in Seurat. Principle component analysis was performed, and the number of PCs used for clustering was determined using the ElbowPlot function. Clusters and markers for clustered were identified using the Louvain algorithm embedded in the FindNeighbours and FindClusters functions, at a resolution of 0.5. UMAP projections were computed using the first ten principal components. Clusters were annotated using the FindAllMarkers function to determine differentially expressed genes for each cluster, then cluster identities were verified using the package SingleR[78]. Heatmaps, violin plots, and UMAP projections were generated using Seurat v4. The FindMarkers function was used to find differentially expressed genes within each cluster between WT and IFNGR1KO conditions. Pathway analysis and plotting of results were performed using the tool fgsea[79]. Volcano plots and bar plots were created using ggplot2. Module scoring of different TAM subsets was done using the package UCell (v.1.3)[80]. Trajectory analysis for CD8+ T cell and macrophage clusters was completed and visualised using the package monocle3 (v.1.0.0)[81,82]. Finally, analysis of cell–cell communication networks and plotting of results were performed using the package CellChat[83]. Data were visualised using Graphpad (V8.4.1, Prism software), GGplot2(v3.3.5) and ggpubr(v0.5.0).

## Analysis of human datasets

Selected TCGA PanCancer Atlas studies were retrieved from cBioPortal[84–86] and queried for cases which contained gene mutations in IFNγ pathway (IFNGR1, IFNGR2, JAK1, JAK2, and STAT1), antigen presentation pathway (HLA-A, HLA-B, HLA-C, B2M, TAP1, and TAP2), or individual genes as indicated. Survival curves for selected cancer types were also retrieved for patient cases which contained the set of IFNγ-pathway mutations versus cases without

such mutations. For endometrial cancer, samples with mutations in the POLE exonuclease domain have been excluded, as this is associated with hypermutated cancers whereby mutations in the IFNγ pathway would not reflect immune pressure. For enrichment scoring of CD8-monocyte signatures in human cancers, normalised STAR gene counts were retrieved from the TCGA for cases of each cancer type and subdivided into control and mutant groups, where mutants were cases which contained confirmed mutations in IFNγ-pathway genes of moderate or severe variant effect predictor (VEP) impact scoring. Signatures of CD8 T cell and monocytes were retrieved and combined to create a CD8-monocyte signature, from the R package consensusTME[87], which has curated cell-type signatures for each tumour type. The function geneSetEnrichment was used to perform single-sample gene set enrichment analysis (ssGSEA) for each tumour type using the custom CD8-monocyte signature. For gene signature analysis of publicly available Visium CytAssist spatial transcriptomic datasets, Loupe browser files for human colon adenocarcinoma (FFPE) and human lung squamous cell carcinoma (FFPE) were downloaded from 10X Genomics and analysed using Loupe Browser 7.0.1. Gene sets were retrieved from reference publications and indicated in Supplementary Data 1, and visualised as log normalised average expression of all features in the gene set.

## Statistics

Unless otherwise noted, all data involving in vivo experiments are pooled from ≥2 separate experiments. Statistical analyses were performed using GraphPad Prism software. Error bars represent standard error of mean (SEM) calculated using Prism. Statistical tests used include non-parametric Mann–Whitney $t$ tests for comparisons between two groups, and two-way ANOVA using Šídák's test for multiple comparisons between multiple two or more groups of data.

## Reporting summary

Further information on research design is available in the Nature Portfolio Reporting Summary linked to this article.

## Data availability

The mouse scRNAseq data generated in this study have been deposited in the GEO database under accession code GSE260972. Datasets retrieved from 10X Genomics are licensed under the Creative Commons Attribution license. All data are included in the Supplementary Information or available from the authors upon reasonable requests, as are unique reagents used in this Article. The raw numbers for charts and graphs are available in the Source Data file whenever possible. Source data are provided with this paper.

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

## Acknowledgements

We would like to thank the Wellcome Trust Centre for Human Genetics for the generation of the sequencing data, J. Webber for the assistance with cell sorting, the dynamic platform and microscopy facility at the Kennedy Institute, and Prof. David Church for discussions on endometrial cancer, and Prof. Irina Udalova and Dr. Linh Nguyen for help with neutrophil depletion. This work was supported by Cancer Research UK (CR-UK) (C5255/A18085 through the Cancer Research UK Oxford Centre, 29549 and DRCPFA-Nov23/100006 to A.G.); the Kennedy Trust for Rheumatology Research (KENN151607 and KENN202112 to A.G.), John Fell Funds (0013739 to A.G.), and Wellcome Trust studentship and Clarendon Scholarship (V.W.C.L.).

## Author contributions

V.W.C.L. and A.G. designed the experiments, analysed the data, and wrote the manuscript. V.W.C.L., G.M., J.M.M., Z.V., and A.K. performed in vivo experiments and processed samples for analysis. V.W.C.L. performed bioinformatic analysis of sequencing and Visium datasets. G.M. prepared imaging samples and performed imaging data collection. G.M. and A.G. performed imaging data analysis. V.W.C.L. and A.G. analysed human datasets. E.W.R. and K.S.M. provided reagents and technical advice for experiments. V.W.C.L., U.G., G.P., V.C., and A.G. contributed to the initial conception of the study.

## Competing interests

The authors declare no competing interests.
