## [Transparent Peer Review file · Nature Communications]

Remodelling of the immune landscape by IFN γ counteracts IFN γ -dependent tumour escape in mouse tumour models

Corresponding Author: Dr Audrey Gerard

Version 0:

Reviewer comments:

Reviewer #1

(Remarks to the Author)

The study presents noteworthy results on how IFN γ , despite tumor cell resistance, continues to remodel the immune landscape, promoting an inflammatory response that enhances tumor control. This finding challenges the traditional understanding that IFN γ resistance in tumor cells leads to diminished immune surveillance and therapy resistance. The work is significant as it elucidates a paradox where tumors that lose IFN γ sensitivity might still be controlled effectively through immune mechanisms, which could significantly influence the design and application of immunotherapies. The concept of leveraging IFN γ insensitivity for therapy, as demonstrated in this study, adds a novel layer to the current understanding and is well-aligned with recent findings but extends them by providing detailed mechanistic insights (Dubrot et al., 2022; Lawson et al., 2020).

The conclusions are supported by the data presented, including detailed scRNAseq analysis and functional validation through in vivo experiments. The study makes a compelling case for the role of IFN γ in maintaining immune surveillance even in the face of tumor cell resistance to IFN γ signaling.

The methodology employed is rigorous and appropriate for addressing the research questions. The use of CRISPR-Cas9 gene editing, scRNAseq, and subsequent functional assays provides a comprehensive approach to understanding the complex dynamics within the tumor microenvironment. The methods section is detailed enough to allow reproducibility, which is crucial for the validation of the results by the broader scientific community.

I do not find any significant flaws in the data analysis or interpretation which would prohibit publication. The data are analyzed with appropriate statistical tools, and the interpretations made by the authors are justified by the results.

Overall, this manuscript makes a significant contribution to the field of cancer immunology by challenging existing paradigms and providing new avenues for therapeutic intervention. I have only a couple of minor comments.

Minor comments:

- 1) Please include more detailed information about the flow cytometry panels and gating strategies in the supplementary materials to ensure that other researchers can precisely replicate the study.
- 2) KRAS mutations appear in similar frequency with all other alterations in endometrial carcinoma (~20%). In addition, why is it that IFN γ mutant endometrial cancers survive better than wt ones (while this is not evident in other tumors)? The authors are advised to provide an explanation for this.

Reviewer #2

(Remarks to the Author)

In their study, Lau et al. suggest that the growth of IFN γ -insensitive tumors (IFN gamma receptor knockout; IFN γ RKO) is controlled by a reprogrammed tumor microenvironment through accumulation of IFN γ as the critical mediator. In this scenario, the increase of inflammatory monocytic and pre-macrophage subsets is proposed as underlying mechanism. The authors describe a reciprocal dependency of CD8+ T cell and CCR2-dependent inflammatory monocyte recruitment. CD8+ T cells were identified as the major source of IFN γ . The study is comprehensive and well controlled. The results are of interest to a broad community of tumor immunologists, but several questions remain.

Questions and comments:

#1 Recent studies reported an important role of CD4+ T cells as instigators of and IFN γ -driven myeloid (monocyte)-dependent control of tumors (also syngeneic melanoma models), addressing also the role of MHC class I or class II deficiency. So, what's the role the CD4+ T cell population in the model presented by Lau et al., e.g. to be addressed by antibody-based depletion.

The authors also discuss a potential role of cross-presentation by the monocyte population to the CD8+ T cells, but CD4+ T cells also play a major role at this cellular interface.

#2 As mentioned by the authors, IFN γ RKO cells still show some basal MHC class I expression and presumably antigen presentation. Given that OVA (SIINFEKL) is an extremely high affinity peptide, what is the role for direct tumor cell recognition? How does IFN γ RKO compare to MHC class I KO or even a double KO?

#3 Even though host IFN γ was important for the control of both WT and IFN γ RKO tumors, Fig. 2G/Fig.S2C show that IFN γ RKO or IFN γ R1-Y445A expressing cells have a growth benefit over WT cells when growing in a shared tumor microenvironment. Is this due to evasion of IFN γ -induced anti-proliferative effects on the tumor cells? Somehow, this finding lacks a mechanistic explanation in view of the other findings.

#4 How do IFN γ RKO tumors actually trigger the re-programming of the tumor microenvironment. Is there a shift in chemokines/cytokines produced by the tumor cells between WT and IFN γ RKO? The role of the tumor cells appears a bit neglected in the study and but may play an important role as well.

Just out of curiosity ...

#5 The authors nicely show that CD8-monocyte populations accumulate around vessels and they support their findings by re-analyzing published spatial transcriptomics data from human tumors. These observations emphasize the importance of the vessel-immune cell interface in tumors, but what is functional role of this accumulation?

#6 Along this line, what is the mechanism by which the monocyte population controls tumor growth? iNOS?

#7 The authors report the CCR2 deficiency triggers a compensatory recruitment of neutrophils into the tumors. This is also an interesting finding, Fig. 5A/B. What is the actual growth kinetic compared to WT/littermate hosts? Is the switch to neutrophil recruitment enhancing tumor growth?

Strengths:

-The study is comprehensive and carefully controlled employing complementary methodologies. Key findings are addressed and supported by different experimental approaches.

-The work indicates an important role for a CD8 T cell – monocyte axis that is centered around tumor vessel mediating control of IFN γ -insensitive tumors.

Limitations:

-The growth kinetics of the model is very fast which may have limitations for adaptive immune responses. All in vivo (tumor growth) experiments are shown as tumor weight analyses at endpoints between day 12 – 14 post inoculation. It is difficult to assess the actual kinetics without seeing tumor growth curves or survival analyses and to compare the different experiments within the study.

Version 1:

Reviewer comments:

Reviewer #1

(Remarks to the Author)

The authors have revised their manuscript and responded point-by-point to my comments. Especially, the exclusion of POLE mutant endometrial cancer patients now makes better sense. I am happy to endorse the publication of this work.

Reviewer #2

(Remarks to the Author)

The authors addressed all my questions by additional experimental data and discussions/clarifications. From my point of view, the manuscript is now suitable for publication.

RESPONSE TO REVIEWER COMMENTS

We would like to thank the reviewers for their time and thorough assessment of our manuscript. Please find below our response to their comments. New text or edits are highlighted in yellow in the manuscript. As a result of those additional experiments and clarifications, we believe the manuscript greatly improved.

Reviewer #1 (Remarks to the Author):

The study presents noteworthy results on how IFN γ , despite tumor cell resistance, continues to remodel the immune landscape, promoting an inflammatory response that enhances tumor control. This finding challenges the traditional understanding that IFN γ resistance in tumor cells leads to diminished immune surveillance and therapy resistance. The work is significant as it elucidates a paradox where tumors that lose IFN γ sensitivity might still be controlled effectively through immune mechanisms, which could significantly influence the design and application of immunotherapies. The concept of leveraging IFN γ insensitivity for therapy, as demonstrated in this study, adds a novel layer to the current understanding and is well-aligned with recent findings but extends them by providing detailed mechanistic insights (Dubrot et al., 2022; Lawson et al., 2020).

The conclusions are supported by the data presented, including detailed scRNAseq analysis and functional validation through in vivo experiments. The study makes a compelling case for the role of IFN γ in maintaining immune surveillance even in the face of tumor cell resistance to IFN γ signaling.

The methodology employed is rigorous and appropriate for addressing the research questions. The use of CRISPR-Cas9 gene editing, scRNAseq, and subsequent functional assays provides a comprehensive approach to understanding the complex dynamics within the tumor microenvironment. The methods section is detailed enough to allow reproducibility, which is crucial for the validation of the results by the broader scientific community.

I do not find any significant flaws in the data analysis or interpretation which would prohibit publication. The data are analyzed with appropriate statistical tools, and the interpretations made by the authors are justified by the results.

Overall, this manuscript makes a significant contribution to the field of cancer immunology by challenging existing paradigms and providing new avenues for therapeutic intervention. I have only a couple of minor comments.

Many thanks for the positive evaluation of our manuscript and the recognition that our data is novel and experiments well performed.

Minor comments:

1) Please include more detailed information about the flow cytometry panels and gating strategies in the supplementary materials to ensure that other researchers can precisely replicate the study.

Thank you for the suggestion which ensures clear and concise communication of our study methods. Flow cytometry panels and gating strategies have now been expanded upon in *Fig.S2H* for the lymphoid panels and *Fig.S4B-C* for the myeloid panel.

2) KRAS mutations appear in similar frequency with all other alterations in endometrial carcinoma (~20%). In addition, why is it that IFN γ mutant endometrial cancers survive better than wt ones (while this is not evident in other tumors)? The authors are advised to provide an explanation for this.

We would like to thank the reviewer for asking us to look into the better survival the endometrial cancer patients when tumours displayed mutations in the IFN γ pathway. We had a discussion with an expert in endometrial cancer at the University of Oxford (Prof. David Church), who pointed out that POLE mutations are prevalent in endometrial cancers, resulting in a widespread mutational landscape. In this case, mutations in the IFN γ pathway might now reflect immune pressure. Because it is known that POLE mutations lead to better patient outcome, we reproduced that analysis by excluding samples with POLE mutations to avoid this confounding factor. In this case, we found no difference in survival between patients with tumours harbouring mutations in the IFN γ pathway versus patients with tumours that do not (*New Fig.1B*). We also reanalysed the frequency of patients exhibiting mutations in the IFN γ pathway when POLE mutations were excluded (*new Fig.1A*). In this case, the rate of mutation in the IFN γ pathway is in the range to the one of melanoma and colorectal cancer. Data is now in line with that of other tumour types.

We clearly stated that we excluded samples with POLE mutations in the Methods section:

For endometrial cancer, samples with mutations in the POLE exonuclease domain have been excluded, as this is associated with hypermutated cancers whereby mutations in the IFN γ pathway would not reflect immune pressure.

And in the legend of Figure 1:

(A) Frequency of alterations in IFNGR1, IFNGR2, JAK1, JAK2, or STAT1 (IFN γ pathway) across cancers in The Cancer Genome Atlas (TCGA), where cases in green represent gene mutations and purple are structural variants of the genes. For endometrial cancer, samples with POLE mutations have been excluded.

(B-E) Comparison of survival curves of endometrial (B, n=462 without and n=50 with mutation in the IFN γ pathway, exclusion of samples with POLE mutations),

Reviewer #2 (Remarks to the Author):

In their study, Lau et al. suggest that the growth of IFN γ -insensitive tumors (IFN gamma receptor knockout; IFN γ RKO) is controlled by a reprogrammed tumor microenvironment through accumulation of IFN γ as the critical mediator. In this scenario, the increase of inflammatory monocyte and pre-macrophage subsets is proposed as underlying mechanism. The authors describe a reciprocal dependency of CD8 $^+$ T cell and CCR2-dependent inflammatory monocyte recruitment. CD8 $^+$ T cells were identified as the major source of IFN γ . The study is comprehensive and well controlled. The results are of interest to a broad community of tumor immunologists, but several questions remain.

Many thanks for assessing our manuscript. We hope that our new data and discussion will answer this reviewer's queries.

Questions and comments:

*#1 Recent studies reported an important role of CD4 $^+$ T cells as instigators of and IFN γ -driven myeloid (monocyte)-dependent control of tumors (also syngeneic melanoma models), addressing also the role of MHC class I or class II deficiency. So, what's the role the CD4 $^+$ T cell population in the model presented by Lau et al., e.g. to be addressed by antibody-based depletion. The authors also discuss a potential role of cross-presentation by the monocyte population to the CD8 $^+$ T cells, but CD4 $^+$ T cells also play a major role at this cellular interface. As suggested by the reviewer, given the role of CD4 T cells in IFN γ -induced myeloid-driven tumour control, we investigated the relative contribution of CD4 T cells to IFN γ production. We are now adding data showing that CD4 T cells are not a major producer of IFN γ in our model (*new Fig.6F*). We believe this is because CD4 T cells make up only 5% of the total*

lymphocytes, as detected in the single-cell experiment, and a majority of them were FoxP3+ (Fig.3A-B and please see Fig.A below). We have added this explanation in the text as:

The few CD4⁺ T cells recruited to tumours were mainly Tregs (Fig.3A-B) and as such, they did not contribute to IFN γ production (Fig.6F).

Fig.A. CD4 T cells present in the B16-OVA model are Tregs. Figure shows CD4 (top), Foxp3 (middle) and CD8a (bottom) expression in lymphoid clusters identified in WT (left) and IFN γ RKO (right) B16-OVA tumours by scRNAseq, related to Fig.3 in the manuscript.

As such, any effect of CD4 depletion would not reflect their contribution to IFN γ -driven macrophage differentiation, but rather would be due to depleting Tregs. Finally, although our model is not CD4 dependent, other models are and therefore, CD4 T cells might contribute to IFN γ -driven myeloid differentiation, as pointed out by this reviewer. We therefore included this in the discussion as such:

IFN γ -mediated macrophage differentiation can also be carried by CD4⁺ T cells in other tumour models⁷¹ and might therefore not always be solely dependent on CD8⁺ T cells.

#2 As mentioned by the authors, IFN γ RKO cells still show some basal MHC class I expression and presumably antigen presentation. Given that OVA (SIINFEKL) is an extremely high affinity peptide, what is the role for direct tumor cell recognition? How does IFN γ RKO compare to MHC class I KO or even a double KO?

As suggested, we investigated the role of MHC-I expression on B16-OVA lines for CD8 T cell expansion. To do so, WT, IFN γ RKO or H2XbKO (H2Db H2Kb double KO) were engrafted in WT mice and the lymphoid infiltrates were quantified at day 10 after engraftment by flow cytometry. This data appears in Fig.S2I-J. Surprisingly, the percentage of overall CD8 T cells and antigen-specific (Tetramer+) CD8 T cells observed in H2XbKO tumours was similar to that of WT tumours. This suggested that the immune microenvironment was responsible for the

infiltration/retention of antigen-specific CD8 T cells. To confirm that CD8 T cells from H2XbKO tumours still received TCR priming from the microenvironment, we grew H2XbKO tumours in Nur77-GFP mice. In these mice, GFP expression is controlled by the Nur77 promoter, and in T cells, Nur77 is induced by TCR but not cytokine priming. We therefore used GFP expression as a read-out for TCR priming by flow cytometry. Antigen-specific (tetramer+) CD8 T cells from H2XbKO tumours express GFP, demonstrating that they are primed in situ (*Fig.S2K*), thereby confirming the involvement of the microenvironment in CD8 T cell priming. This now appears in the text as:

Interestingly, OVA-specific CD8⁺ T cell infiltration is similar between WT and H2-Kb/Db KO tumours (*Fig.S2I-J*), highlighting the importance of the tumour microenvironment for recruiting/maintaining antigen-specific CD8⁺ T cells at the tumour site. To confirm that CD8⁺ T cells were still primed by their TCR in H2-Kb/Db KO tumours, we used the Nur77-GFP reporter mice, whereby GFP expression is induced by TCR, but not cytokine signalling in T cells⁴⁴. OVA-specific CD8⁺ T cells expressed GFP to a greater extent than the tetramer-negative CD8⁺ T cells (*Fig.S2K*), confirming that the microenvironment is critically important to support CD8⁺ T cells priming and most likely their maintenance/expansion. Furthermore, increased antigen-specific CD8⁺ T cell infiltration following IFN γ R depletion has been observed in other model antigens and cell lines^{15, 45}, suggesting that this is not an artefact of OVA-expressing tumours.

#3 Even though host IFN γ was important for the control of both WT and IFN γ RKO tumors, Fig. 2G/Fig.S2C show that IFN γ RKO or IFN γ R1-Y445A expressing cells have a growth benefit over WT cells when growing in a shared tumor microenvironment. Is this due to evasion of IFN γ -induced anti-proliferative effects on the tumor cells? Somehow, this finding lacks a mechanistic explanation in view of the other findings.

As suggested, we investigated the mechanism by which IFN γ RKO tumours take over their WT counterpart when in competition. We used an in vitro system where we mixed WT and IFN γ RKO tumour cells and tested the direct function of IFN γ on inhibiting proliferation and the indirect function of IFN γ on T cell killing. This now appears in new *Fig.2J-L*. IFN γ treatment of the admix resulted in lower proliferation of WT compared to IFN γ RKO tumours as assessed by KI67 staining (*Fig.2J*), which resulted in an increased number of IFN γ RKO tumours compared to WT tumours (*Fig.2K*). To investigate T cell killing, tumours were treated with a low dose of IFN γ to drive MHC-I upregulation, and pre-activated OTI T cells were added to the admix tumours. We observed a small, but consistent, increased Annexin V staining of WT tumours compared to IFN γ RKO tumours (*Fig.2L*), suggesting WT tumours are slightly more sensitive to T cell killing than IFN γ RKO tumours. We edited the text as:

To characterise how CD8⁺ T cells and IFN γ led to the selection of IFN γ RKO tumour cells in our admix setup, we first analysed whether IFN γ had a direct cytostatic effect on WT tumours. To do so, we admixed WT and IFN γ RKO B16-OVA tumour cells *in vitro* and added IFN γ to the culture. The proliferation of IFN γ RKO tumour cells, as assessed by KI67 staining, was greater than the one of WT tumour cells (*Fig.2J*), demonstrating that IFN γ inhibits tumour cell proliferation, which contributed to the selection of IFN γ RKO over WT tumour cells (*Fig.2K*). To test whether CD8⁺ T cells could also induce IFN γ RKO tumour cell selection through preferential killing of WT tumour cells, we ad-mixed WT and IFN γ RKO tumour cells and analysed cell death induced by OVA-specific OTI CD8⁺ T cells using Annexin V staining. We observed a slight but consistent preferential killing of WT over IFN γ RKO tumour cells by OTI CD8⁺ T cells (*Fig.2L*), consistent with the increase of MHC-I observed in the WT but not IFN γ RKO tumour cells following IFN γ treatment (*Fig.2C*). Overall, we concluded that a direct effect of IFN γ on proliferation and indirect effect on MHC-I expression and subsequent killing by CD8⁺ T cells contribute to the selection of WT over IFN γ RKO tumour cells in our ad-mix model.

#4 How do IFN γ RKO tumors actually trigger the re-programming of the tumor microenvironment. Is there a shift in chemokines/cytokines produced by the tumor cells between WT and IFN γ RKO? The role of the tumor cells appears a bit neglected in the study and but may play an important role as well.

We agree that we have not fully characterised the direct relevance of IFN γ signalling on tumours for reprogramming the immune environment. To investigate the direct role of IFN γ in cytokine/chemokine production by tumour cells, we treated WT B16-OVA with IFN γ for the indicated period and assessed a panel of inflammatory cytokines using a legendplex kit (IFN γ , IL-10, CCL4 (MIP-1 β), IFN α , CXCL9 (MIG), CXCL10 (IP-10), TNF- α , IL-6, VEGF, IL-4, CCL3 (MIP-1 α), CCL2 (MCP-1), and GM-CSF). Only CXCL10 was found to be induced by IFN γ (Fig.S3E). The concentration of CXCL10 in supernatants from IFN γ RKO tumours was slightly lower than in WT tumours (Fig.S3F). The receptor of CXCL10, CXCR3, is predominantly expressed in lymphocytes (new Fig.S3G). Given that the recruitment of lymphoid cells still occurs in IFN γ RKO tumours, the slight decrease in CXCL10 is unlikely to explain the differences in microenvironment and milieu. However, in an admix tumour, it is possible that the lack of expression of CXCL10 by IFN γ RKO tumours might contribute to T cells preferentially targeting WT tumours. This now appears in the text as:

To investigate whether tumours themselves might drive changes in the microenvironment, we treated WT B16-OVA with IFN γ for the indicated period and assessed the expression of a panel of cytokines, for which only CXCL10 was found to be induced by IFN γ (Fig.S3E). The concentration of CXCL10 in supernatants from IFN γ RKO tumours was slightly lower than in WT tumours (Fig.S3F). The receptor of CXCL10, CXCR3, is predominantly expressed in lymphocytes (Fig.S3G). Given that the recruitment of lymphoid cells still occurs in IFN γ RKO tumours, the slight decrease in CXCL10 is unlikely to explain differences in microenvironment and milieu observed between WT and IFN γ RKO tumours. However, in an admix tumour, the lack of CXCL10 expression by IFN γ RKO tumours might contribute to T cells preferentially targeting and killing WT tumours. Given that we did not detect major differences in the secretion of cytokine/chemokine between tumour types, we hypothesised that immune cells were driving differences in the tumour milieu, and employed CellChat on the immune scRNAseq dataset as a tool for dissecting soluble signals and cell-cell communications occurring in tumours.

Just out of curiosity ...

#5 The authors nicely show that CD8-monocyte populations accumulate around vessels and they support their findings by re-analyzing published spatial transcriptomics data from human tumors. These observations emphasize the importance of the vessel-immune cell interface in tumors, but what is functional role of this accumulation?

We highlight that CD8 T cells must produce IFN γ in the tumour, but it is not known if where the IFN γ is produced spatially makes a difference. Using GREAT mice, we provide evidence that CD8 T cells mainly produce IFN γ in clusters close to blood vessels. As a result, it's likely that the IFN γ response initiates near vessels, and resulting local inflammation is strategically positioned, providing differentiation to monocytes as soon as they enter the tumour. While this might not be the only function of this vessel-immune interface, it is reasonable to hypothesise that this is an important role, as inhibiting both IFN γ production or ablating CD8 T cells abolishes monocyte recruitment and skewing in IFN γ -insensitive tumours. This now appears in the result section as:

Thus, while monocytes need CD8⁺ T cells for their recruitment regardless of the tumour type, inhibition of IFN γ sensing in tumours leads to an increase in the ability of CD8⁺ T cells to recruit

monocytes. These findings are consistent with studies which describe the role of pro-inflammatory cytokines such as IFN γ in enabling leukocyte adhesion and transendothelial migration through integrin^{62, 63} or MHC class II upregulation⁶⁴. In IFN γ RKO tumours, higher IFN γ levels may increase adhesion of lymphocytes and monocytes to intra-tumoural endothelium, which we observe as a quantifiable increase in these cell-cell interactions. Consistent with this, IFN γ , highlighted using GREAT mice, was produced by CD8⁺ T-cells and occurred primarily around blood vessels and in close proximity to some of the infiltrating monocytes (Fig.7D-E). This suggested that the strategic positioning of IFN γ -producing CD8⁺ T-cells either supported monocyte recruitment, and/or allowed for monocytes to receive differentiation signals as soon as they entered tumours. Although monocytes are not necessarily directly in contact with IFN γ -producing cells, IFN γ spread can reach between 3 to 30 cells depending on models and T cell density^{65, 66, 67}, which is in line with the close proximity between IFN γ -producing cells and monocytes we observed (Fig.7E).

Overall, our data demonstrate that the CD8/monocyte cross-talk occurs around vessels in IFN γ -insensitive tumours, where CD8 T cells recruit monocytes and skew their differentiation.

And in the discussion as:

We found that IFN γ -production by CD8⁺ T cells mostly occurred around blood vessels in tumours. Perivascular immune niches that contain CD8⁺ T cells, Dendritic Cells and activated macrophages have been correlated with anti-tumour immunity⁷⁵, and likely supports IFN γ production in this strategic perivascular area, inducing rapid differentiation of monocytes into macrophages.

#6 Along this line, what is the mechanism by which the monocyte population controls tumor growth? iNOS?

To assess whether iNOS is important to control IFN γ RKO tumours, we inhibited iNOS using the inhibitor L-NAME administered in drinking water, and measured tumour growth. Interestingly, L-NAME increased the growth of IFN γ RKO, but not WT tumours. This data appears in Fig.5C and S5D. This is in line with the fact that IFN γ RKO tumours exhibit an increased proportion of NOS2⁺ macrophages compared to WT tumours, and suggests that macrophage control IFN γ RKO tumours at least through iNOS. This appears in the text as:

Because IFN γ RKO tumours are characterised by an increase in monocyte-derived NOS2⁺ macrophages (Fig.4A-C), we hypothesised that macrophages control IFN γ RKO tumour growth through NOS2-induced nitric oxide (NO), which can exert anti-tumour effects^{57, 58}. To test this, we treated mice engrafted with WT or IFN γ RKO tumours with the NOS2 (iNOS) inhibitor L-NAME. iNOS inhibition increased tumour growth of IFN γ RKO tumours (Fig.5C, S5D). iNOS-induced tumour control was less effective in WT tumours (Fig.5C, S5D), consistent with the fact that less NOS2⁺ macrophages are present in WT compared to IFN γ RKO tumours (Fig.4A-C). Overall, we concluded that monocyte-derived macrophages control growth of IFN γ RKO tumours, in part through iNOS.

#7 The authors report the CCR2 deficiency triggers a compensatory recruitment of neutrophils into the tumors. This is also an interesting finding, Fig. 5A/B. What is the actual growth kinetic compared to WT/littermate hosts? Is the switch to neutrophil recruitment enhancing tumour growth?

To address the role of neutrophils in the control IFN γ RKO tumours, we used a protocol adapted from Boivin, G. et al., Nat Comms 2020, to deplete neutrophils (Fig.S5B). Neutrophil depletion did not affect tumour growth of IFN γ RKO tumours (Fig.S5C). We concluded that in this model, neutrophils had a marginal effect on tumour growth. It should also be noted the apparent increase in neutrophils, reported as a proportion, might also be partially driven by the loss of macrophage and T cells. This appears in the text as:

Neutrophil depletion in CCR2KO mice engrafted with IFN γ RKO tumours had no effect on tumour growth (Fig.S5B-C), suggesting that the lack of monocyte recruitment, rather than the

increase in neutrophils, was responsible for enhanced IFN γ RKO tumour growth in CCR2KO mice.

We are also providing the growth kinetics in CCR2KO mice in *Fig.S5A*.

Strengths:

-The study is comprehensive and carefully controlled employing complementary methodologies. Key findings are addressed and supported by different experimental approaches.

-The work indicates an important role for a CD8 T cell – monocyte axis that is centered around tumor vessel mediating control of IFN γ -insensitive tumors.

Many thanks for highlighting the strengths of our study.

Limitations:

-The growth kinetics of the model is very fast which may have limitations for adaptive immune responses. All in vivo (tumor growth) experiments are shown as tumor weight analyses at endpoints between day 12 – 14 post inoculation. It is difficult to assess the actual kinetics without seeing tumor growth curves or survival analyses and to compare the different experiments within the study.

The reviewer is right that this model is fast growing. To provide a better representation of tumour growth, we have added the growth curves over time (WT vs IFN γ RKO vs admix tumours: *Fig.S2B*; WT vs CD8KO mice: *Fig.S2G*, CCR2KO mice: *Fig.S5A*, neutrophil depletion in CCR2KO mice (IFN γ RKO B16-OVA): *Fig.S5C*, Inos inhibition: *Fig.S5D*, WT vs IFN γ KO mice: *Fig.S6A*).

RESPONSE TO REVIEWERS' COMMENTS

Reviewer #1 (Remarks to the Author):

The authors have revised their manuscript and responded point-by-point to my comments. Especially, the exclusion of POLE mutant endometrial cancer patients now makes better sense. I am happy to endorse the publication of this work.

Many thanks for endorsing our manuscript

Reviewer #2 (Remarks to the Author):

The authors addressed all my questions by additional experimental data and discussions/clarifications. From my point of view, the manuscript is now suitable for publication.

Many thanks for endorsing our manuscript